# Structural insights into the role of GTPBP10 in the RNA maturation of the mitoribosome

Thu Giang Nguyen[1], Christina Ritter ®[1] & Eva Kummer ®[1] ✉

Mitochondria contain their own genetic information and a dedicated translation system to express it. The mitochondrial ribosome is assembled from mitochondrial-encoded RNA and nuclear-encoded ribosomal proteins. Assembly is coordinated in the mitochondrial matrix by biogenesis factors that transiently associate with the maturing particle. Here, we present a structural snapshot of a large mitoribosomal subunit assembly intermediate containing 7 biogenesis factors including the GTPases GTPBP7 and GTPBP10. Our structure illustrates how GTPBP10 aids the folding of the ribosomal RNA during the biogenesis process, how this process is related to bacterial ribosome biogenesis, and why mitochondria require two biogenesis factors in contrast to only one in bacteria.

Mitochondria are eukaryotic organelles with a central role in cell metabolism, regulation of apoptosis, cell differentiation, and innate immunity. They contain their own genetic material that encodes for approximately a dozen essential components of the respiratory chain complexes in humans. To express their genome, mitochondria maintain a distinct translation apparatus that differs from its cytosolic counterpart and the bacterial ancestor. Especially, the architecture and composition of the mitochondrial ribosome (mitoribosome) has changed representing an unusually protein-rich ribosome with reduced RNA elements in humans[1,2]. The human mitochondrial ribosome is assembled in the mitochondrial matrix from ribosomal RNA (rRNA) encoded on the mitochondrial DNA and ribosomal proteins of nuclear origin. After their syntheses in the cytosol, the mitochondrial ribosomal proteins are imported into the organelle and associate with the rRNA in a coordinated and hierarchical fashion[3,4].

The assembly of the mitoribosomal small and large subunits (mtSSU and mtLSU, respectively) is assisted by dedicated biogenesis factors that temporarily bind to the maturing particles to establish the timely and coordinated incorporation of ribosomal proteins as well as the folding and modification of rRNA. Biogenesis factors are both, conserved or mitochondria-specific and include methyltransferases and pseudouridine synthases, helicases, GTPases, and other proteins[3–5]. Defects in ribosome biogenesis reduce mitochondrial protein synthesis and cause severe human diseases including encephalomyopathy, optic neuropathy, and spastic paraplegia[6].

Recently, several single particle cryo-electron microscopy (cryo-EM) structures of biogenesis intermediates of the human mtLSU have provided a detailed view of the later stages of the assembly process, where the outer shell of the particle has already formed but the subunit interface is still immature. These structures have been vital to understand the order, in which the single components are assembled into the final functional mitoribosome, as well as the positioning, function, and interplay of biogenesis factors in this process[7–12]. A remarkable feature of the assembly of the mtLSU is that the catalytic site of the ribosome—the peptidyl transferase center (PTC)—folds last to ensure that only properly assembled particles enter the translation process[7]. This quality control checkpoint is highly conserved and a common feature in the biogenesis of the ribosome in all translation systems[3]. Despite the wealth of mtLSU maturation snapshots, additional key intermediates have so far escaped structural elucidation.

Here, we present the structure of a late-stage maturation intermediate of the human mtLSU at an overall resolution of 3.03 Å. Our complex contains the NSUN4-MTERF4 dimer, the MALSU1-L0R8F8-mtACP module, and the GTPases GTPBP7 and GTPBP10 at the immature, mitoribosomal large subunit interface. GTPBP7 (also termed MTG1) was shown to be a homolog of bacterial RbgA and its depletion impairs mitochondrial translation and assembly of the respiratory chain due to maturation defects of the mtLSU[13]. GTPBP10 (also termed OBGH2) is one of two mitochondrial homologs of the bacterial GTPase ObgE with GTPBP5 (also termed OBGH1 or MTG2) being the other one. Biochemical and cell biological work has previously identified GTPBP5

[1]Novo Nordisk Foundation Center for Protein Research, University of Copenhagen, Blegdamsvej 3B, 2200 Copenhagen, Denmark.
✉e-mail: eva.kummer@cpr.ku.dk

and GTPBP10 as GTPases that bind to late mtLSU maturation intermediates[14–17]. Loss of either leads to reduced levels of 16S rRNA, a reduction in mitochondrial protein synthesis and consequently cell growth. Both, GTPBP5 and GTPBP10 can complement for loss of ObgE in *E. coli*[18], but have nonoverlapping, essential functions in human mitoribosome biogenesis[14–17]. Intriguingly, GTPBP10 and GTPBP5 have largely similar predicted folds and differ mostly in the sequence of their Obg domains. How these differences in GTPBP10 and GTPBP5 convey distinct molecular functions during the mitoribosome assembly process remains unknown. GTPBP10 has been assigned to a low-resolution EM map previously[10]. However, in our EM reconstruction, we discover it in a different conformation on the mtLSU and in contact with the GTPase GTPBP7. Our data permit to rationalize its role in active site maturation via H89 positioning. Our structure also shows why the mitoribosomal maturation process requires two distinct homologs of the highly conserved bacterial maturation factor ObgE.

## Results

### Isolation of assembly intermediates of the human mtLSU

We initially intended to trap mitoribosomal translation termination complexes with a catalytic inactive mutant of the termination factor mtRF1, which has been shown to decode the non-canonical stop codons AGG and AGA in human mitochondria[19–21]. To this end, we purified mitoribosomes in the presence of the non-hydrolysable nucleotide analog GMPPNP from HEK293-6E cell, in which we had transiently overexpressed mtRF1-AAG-3xFLAG. The mitoribosomal pool was analyzed by single particle cryo-EM. Although we did not identify a particle subset containing mtRF1, we computationally isolated the mitochondrial ribosome in complex with translation elongation factor mtEFG1, the initiating 55S mitoribosome containing mtIF2, a contamination of 80S cytoplasmic ribosomal particles with elongation factor eEF2 bound, as well as 3 distinct mtLSU maturation intermediates (Fig. 1a). The mtLSU maturation intermediates contained either the biogenesis factors NSUN4-MTERF4, and the MALSU1-L0R8F8-mtACP module in the presence of GTPBP5 (intermediate 3), NSUN4-MTERF4 and the MALSU1-L0R8F8-mtACP module in the presence of GTPBP7 (intermediate 2), or NSUN4-MTERF4 and the MALSU1-L0R8F8-mtACP module in the presence of GTPBP7 and GTPBP10 (intermediate 1) (Fig. 1a). As the high-resolution structures of all but one of these complexes have been described elsewhere[8,9,12,22–28], we decided to focus our attention on the maturation intermediate containing the GTPases GTPBP7 and GTPBP10.

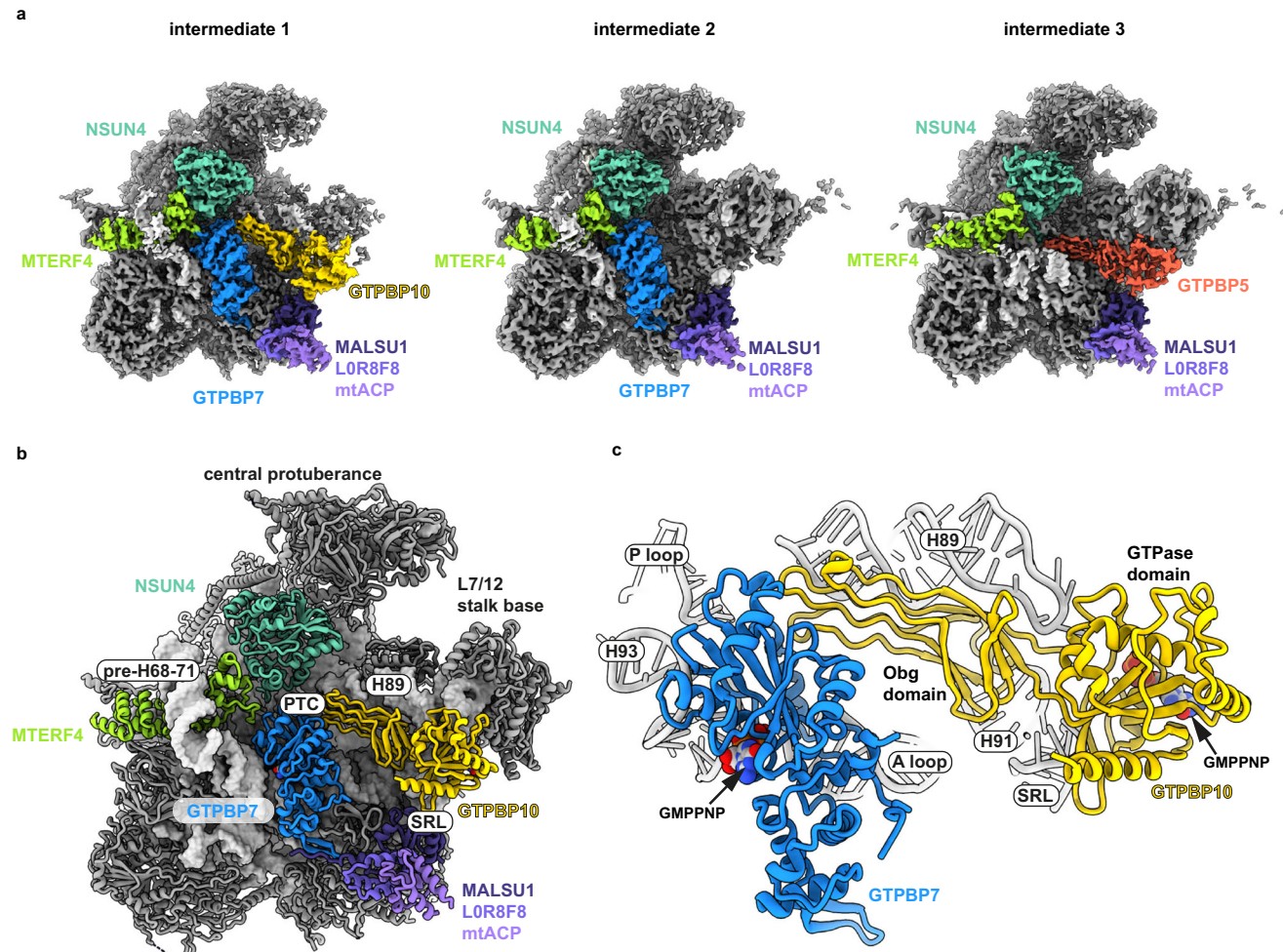

**Fig. 1 | Overview of mtLSU biogenesis intermediates. a** Cryo-EM densities of the isolated mtLSU maturation intermediates with the associated biogenesis factors colored. The colors of the protein names and respective densities are matched. **b** Overview of the structural model of intermediate 1 containing the GTPases GTPBP10 (yellow), GTPBP7 (blue), the module of MALSU1, L0R8F8, mtACP (shades of violet), and the dimer of MTERF4 (yellowgreen) and NSUN4 (cyan). The ribosomal RNA is shown as surface representation whereas all proteins are shown as cartoons. Prominent ribosomal elements are labeled including the sarcin-ricin loop (SRL), the peptidyl-transferase center (PTC), and ribosomal rRNA helix H89 (H89). **c** GTPases GTPBP10 and GTPBP7 are shown in isolation with important ribosomal RNA elements highlighted. Both GTPases likely contain the non-hydrolysable nucleotide analog Guanosine-5′-[(β,γ)-imido]triphosphate (GMPPNP) in their active sites.

## A network of maturation factors at the mtLSU interface

The catalytic core of the mitoribosome is mostly composed of RNA elements and folds last in the maturation process[29]. Key RNA elements encompass the PTC with A, P, and PTC loops, and the GTPase activating center (GAC) that consists of RNA and protein elements around the L7/12 stalk base including the sarcin-ricin loop (SRL), and protein uL11m. While the GAC serves the binding and activation of translational GTPases, the PTC is the site of peptide bond formation where A and P loops facilitate binding of the acceptor ends of A and P site tRNAs, respectively[30–33].

In our structure, we find the immature PTC surrounded by a heterodimer of the methyltransferase NSUN4 and the RNA-binding protein MTERF4, as well as GTPases GTPBP7 and GTPBP10 (Fig. 1b). In addition, a heterotrimer of biogenesis and anti-association factors MALSU1, L0R8F8, and mitochondrial acyl carrier protein (mtACP) interacts with the SRL, uL14m and bL19m as seen in earlier mtLSU maturation intermediates and a ribosome rescue complex[7–12,34]. Similar to previous observations, the MTERF4-NSUN4 dimer contains its cofactor S-adenosyl-methionine (SAM) although in a position incompatible with methylation of 16S rRNA (Supplementary Fig. 1a). The heterodimer binds the double stranded rRNA elements of the pre-H68-71 region as evidenced by continuous EM density that we and others can trace from H67[8,11]. Both, NSUN4−MTERF4 and MALSU1−L0R8F8−mtACP bind early in the mtLSU maturation process and persist over multiple maturation steps.

In contrast, GTPases engage with the maturing ribosome mostly at very specific stages of the maturation process. GTPBP10 is one of the two mitochondrial homologs of bacterial ObgE. It consists of an N-terminal Obg domain and a C-terminal GTPase domain (Fig. 1c). In our structure, the protein is bound in a crevice reaching from the GAC to the immature, catalytic PTC (Fig. 1b). The GTPase domain is located between the L7/12 stalk base and the SRL in a catalytically competent orientation. The Obg domain interacts extensively with the ribosomal RNA helix H89, which is trapped in a lifted-out conformation in comparison to the mature mtLSU (Supplementary Fig. 1c−e). GTPBP10 contacts the second GTPase GTPBP7 close to the PTC, where GTPBP7 in turn binds the heterodimer of NSUN4-MTERF4 (Fig. 1b, c). GTPBP7 is a homolog of bacterial biogenesis factor RbgA, which facilitates incorporation of bL36 in bacteria[35]. It adopts a position on the maturing mtLSU earlier observed only in the absence of the methyltransferase MRM2 as well as in kinetoplastid mtLSU biogenesis[12,35–38]. GTPBP7 has been proposed to monitor the 2′-O-ribose methylation status of the highly conserved nucleotides U3039 and possibly also G3040 in the A loop, which are critical for biogenesis and catalytic activity of the ribosome[12]. Our data suggest that both GTPBP10 and GTPBP7 harbor the non-hydrolysable nucleotide analog Guanosine-5′-[(β,γ)-imido]triphosphate (GMPPNP) bound in their active sites (Supplementary Fig. 2). In accordance with earlier biochemical data, our maturation intermediate already contains all ribosomal proteins of the mtLSU except bL36m, whose binding site gets only accessible once H89 adopts a mature conformation[14].

A low resolution cryo-EM structure has earlier claimed to have identified GTPBP10 at the maturing mtLSU interface, but has placed it in a non-canonical conformation with the Obg instead of the GTPase domain contacting the SRL (Supplementary Fig. 3)[10]. Also, the rRNA helix H89 is shown in a distinct conformation in that model and displays major clashes with the putative GTPBP10. Overall, the resolution of the previous reconstruction can be considered too low to assign the density with certainty to GTPBP10. In our EM density map, we now unambiguously identify GTPBP10 that is in contact with other maturation factors and engages in extensive interactions with H89. It moreover associates with the catalytic center of its GTPase domain to the SRL indicating our conformation displays a catalytically competent state (Fig. 1b, c, Supplementary Fig. 3).

## The role of GTPBP10 in H89 maturation

Recently, cryo-EM structures of the homologous GTPase ObgE have been solved on the native pre-50S and mature 50S LSU from *E. coli*[39]. Structures and biochemical evidence provide mechanistic insight how ObgE aids the organization of the LSU catalytic center during bacterial ribosome biogenesis. The structures highlight a role of ObgE in folding of the 23S rRNA helix H89, allowing incorporation of ribosomal proteins uL16 and bL36 in a subsequent assembly step (Supplementary Fig. 4a−c).

In human mitochondria, two homologs of the essential ObgE exist, namely GTPBP5 and GTPBP10. Structural insights into GTPBP5 bound to an mtLSU intermediate indicate that H89 has already adopted a fully mature conformation (Fig. 2a, Supplementary Fig. 1e). In contrast, our structure shows that the GTPBP10-bound state is characterized by a large displacement of H89 indicating that GTPBP10 acts prior to GTPBP5 during mitochondrial ribosome biogenesis (Fig. 2b, Supplementary Fig. 1d). While H89 is almost completely folded and most of its base pairing has been established in the GTPBP10 maturation intermediate, the H89 base including the PTC loop is entirely disordered to allow the RNA to adopt this lifted conformation. Moreover, the tip of H89 is stretched by clamping between the L7/12 stalk base and the Obg domain of GTPBP10 (Fig. 2c). GTPBP10 fixes H89 in this conformation mostly via electrostatic interactions of the Obg domain with the sugar-phosphate backbone of H89 (Fig. 2d) and via insertion of an alpha-helical element (R52-R66) of the Obg domain between the tip of rRNA helix H89 and the underlying H91 (Fig. 2e). In addition, the GTPBP10 GTPase domain is situated on the SRL and its Obg domain establishes extensive contacts to GTPBP7 (Fig. 1c and Fig. 2b, Supplementary Fig. 5a−d). These interactions position GTPBP10 on the maturing particle allowing it to act as a 'door stopper' for H89.

Overall, GTPBP10 interactions with GTPBP7, H91 and the H89 RNA backbone make it sterically impossible for H89 to slip into its final position. GTPBP10 release is necessary to resolve this steric block and allow accommodation of H89 in its binding pocket on the mtLSU. Despite its low resolution in our EM density, we were able to fit the L7/12 stalk base as a rigid body and find that it is shifted outwards in comparison to the mature, translation factor-free mtLSU and the GTPBP5-bound pre-39S (Fig. 2f−h, Supplementary Fig. 5e). Dislocation of the stalk base is due to the stretched conformation of the H89 tip, which pushes against rRNA helices H43 and H44 (Fig. 2h). H89 accommodation will allow relocation of the stalk base and uL11m. uL11m will subsequently collide with a GTPBP10-specific insertion in its GTPase domain (around residues 240-247) indicating that H89 accommodation is necessarily connected to release of GTPBP10 from the GAC (Fig. 2f). The dislocation of the stalk base leaves the binding site for bL36m still inaccessible in intermediate 1, in accordance with biochemical data[14]. Although we can identify density for uL16m, it is still rather loosely attached to the immature H89 (Supplementary Fig. 5f). Overall, our data confirm that H89 folding and accommodation via GTPBP10 are necessary for sequential incorporation of uL16m and later also bL36m into the mtLSU.

## Comparison of GTPBP10 to GTPBP5 and bacterial ObgE

GTPBP5 and GTPBP10 have nonoverlapping, essential functions in human mitoribosome biogenesis[14–17]. Both proteins have largely similar folds but display differences in their Obg domains (Fig. 3a, b). Our structure now shows that GTPBP10 contains truncations especially in loop 1 and loop 3, and an alpha-helical insertion between beta strands 2 and 3 of the Obg domain. These structural differences explain the distinct roles of both GTPases as the extended loops of GTPBP5 are sterically incompatible with GTPBP7 interaction (Fig. 3d). In contrast, the shortened loops as well as the alpha-helical element allow GTPBP10 to intimately nestle onto GTPBP7 and to interact extensively with H89 to trap it in a lifted conformation (Fig. 3d, Supplementary Fig. 6). The

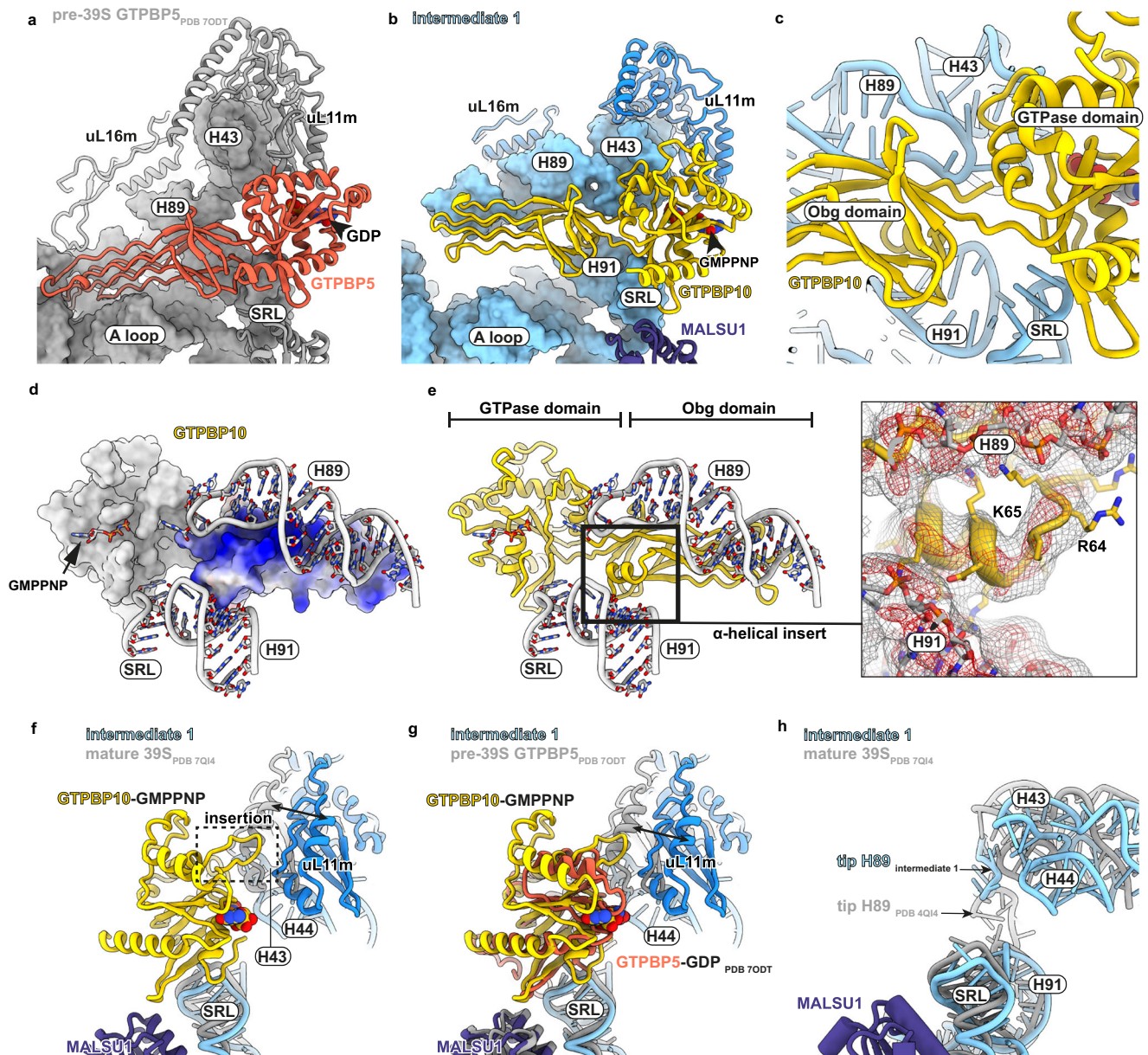

**Fig. 2 | Maturation of H89 by GTPBP10. a**, **b** Overview of the position of GTPBP5 and GTPBP10 and the mtLSU interface. Notable is the difference between the location of H89 and the L7/12 stalk base in the GTPase activating center (GAC) between both intermediates. Important ribosomal RNA elements and mitoribosomal protein uL16m have been labeled. **c** Close-up of the tip of H89 and its surrounding elements. The tip is located between the RNA helices H43 and H44 of the GAC, H91, and the junction between Obg and GTPase domains of GTPBP10. **d** Surface representation of GTPBP10 colored according to coulombic electrostatic potential using default settings (blue = positive (10), red = negative (−10)) in Chi-meraX. Ribosomal RNA helices H89, H91 and the sarcin-ricin loop (SRL) are shown for reference. **e** GTPBP10 structural model shown in the same orientation as in panel (**d**). Organization and positioning of GTPase and Obg domains are high-lighted and the non-hydrolysable nucleotide analog GMPPNP is modeled in its binding site. Important surrounding ribosomal RNA elements are shown. The GTPBP10-specific alpha helical element that aids H89 positioning is boxed and shown in more detail on the right. The corresponding sharpened and 5-times supersampled EM density has been included as isomesh at two thresholds (red = 3.2 σ, gray = 2.3 σ). **f**, **g** Position of the SRL and the L7/12 stalk base in intermediate 1 in comparison to the mature mtLSU (PDB 7QI4) or the GTPBP5-bound intermediate (PDB 7ODT). For intermediate 1, rRNA of is shown in light blue and its ribosomal protein uL11m in blue while the same elements are shown in gray for the mature mitoribosome (**f**) or the GTPBP5-bound intermediate (**g**). The displacement of the stalk base is highlighted with respect to uL11m using a black arrow. GTPBP10 contains an insertion (boxed in f) in its GTPase domain that would clash with a stalk base conformation present in the mature mitoribosome. GTPBP5 lacks this inser-tion (shown in g). **h** Same as f but the ribosomal protein uL11m and GTPBP10 have been removed for clarity. The location of the tip of H89 is moreover highlighted.

extensive interaction of the Obg domain with the RNA backbone is also reflected in a higher local density of positively charged amino acid side chains in GTPBP10 in comparison to GTPBP5 (Supplementary Fig. 6a–c). However, the extended loops of GTPBP5 allow it to reach the unstructured base of H89 to trigger its folding (Fig. 3b, Supple-mentary Fig. 6d, e). This indicates that GTPBP10 catalyses accom-modation of H89 into its binding site on the mtLSU, while GTPBP5

completes H89 and PTC maturation only in a later step as it promotes folding of the H89 base including the PTC loop. Curiously, bacterial ObgE is a chimera of both mitochondrial versions, as it contains elongated loops 1 and 3 as well as the alpha-helical element in its Obg domain explaining why it can complete both, accommodation and folding of H89 (Fig. 3c, Supplementary Fig. 6c). Moreover, while ribosomal protein uL16 joins the LSU in bacteria only when ObgE has

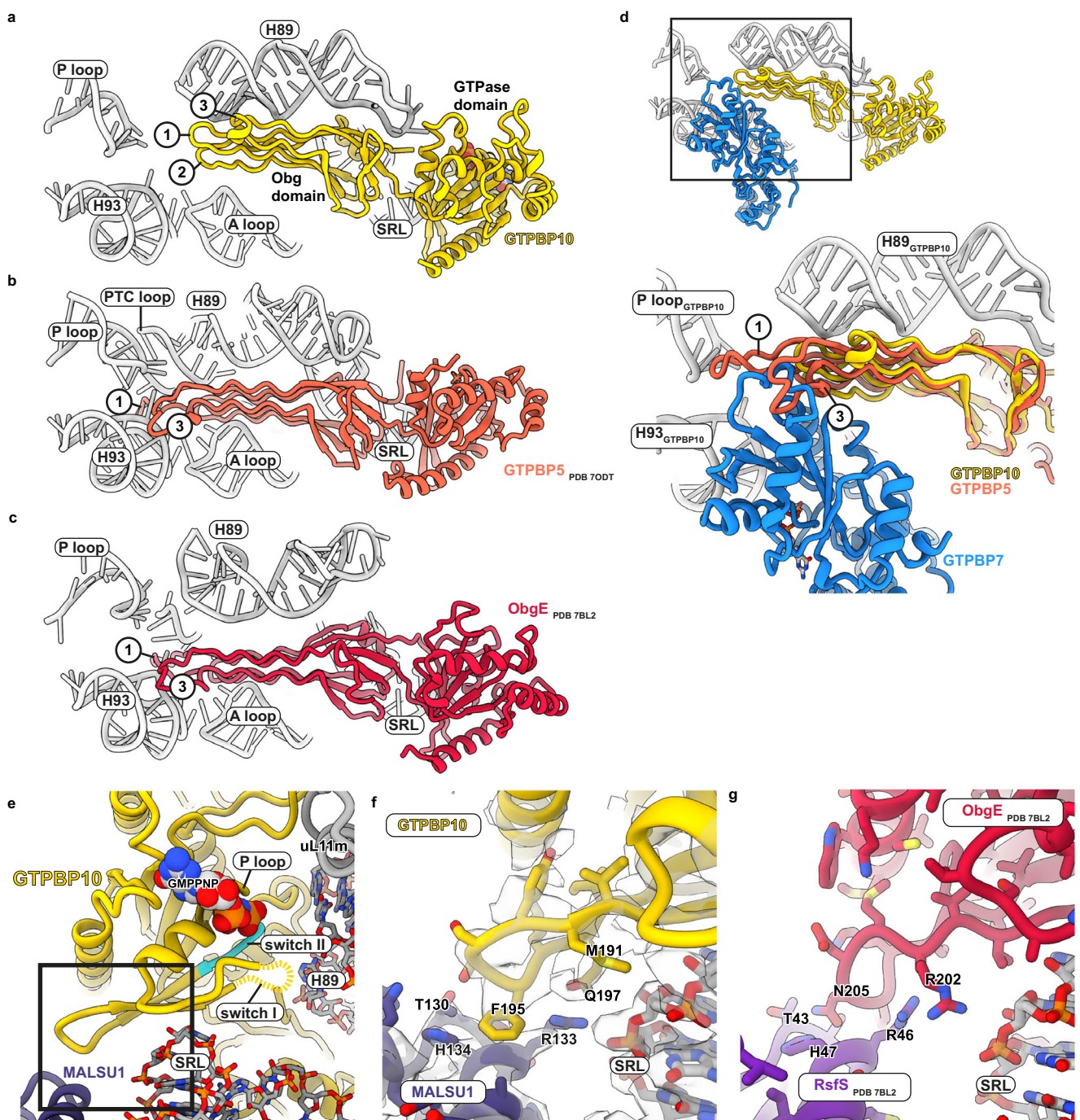

**Fig. 3 | Comparison of mitochondrial GTPBP10, GTPBP5, and bacterial ObgE.**
**a**–**c** A comparison of the mitochondrial ObgE homologs GTPBP10 and GTPBP5 with its bacterial counterpart is shown. For comparison the ribosomal RNA of the large subunit from intermediate 1, PDB 7ODT (GTPBP5), and PDB 7BL2 (ObgE) has been superimposed and conserved ribosomal elements are highlighted in the images. It is apparent that the proteins diverge in the length of their Obg domain loops 1 and 3, which have been labeled with numbers in the images. SRL = sarcin-ricin loop, PTC = peptidyl transferase center (**d**) Superposition of the Obg domain of GTPBP5 (PDB 7ODT) onto the Obg domain of GTPBP10 shows that GTPBP5 is sterically incompatible with GTPBP7 in intermediate 1. **e** Overview of the location of the

GTPase domain and the bound nucleotide in the context of ribosomal RNA helix H89, the ribosomal stalk base containing uL11m, the sarcin-ricin loop (SRL), and biogenesis factor MALSU1. Switch II containing the catalytic Walker B motif (DxxG) is colored in cyan highlighted. The region detailed in (**g**) is boxed. **f**, **g** Contact sites between GTPBP10 and MALSU1 as well as the bacterial ObgE with RsfS (PDB 7BL2) are shown. The sharpened, supersampled EM density for intermediate 1 is displayed as semi-transparent surface. The loop likely aids to anchor the GTPBP10 GTPase domain via MALSU1 on SRL as it was one of the best-resolved regions within the GTPase domain.

almost completed H89 accommodation, it is present in mitoribosomes already when H89 is far from its final position (Supplementary Fig. 4). This is possible because the bacterial rRNA element H38 is significantly shortened in mitoribosomes and consequently the binding site for uL16m becomes accessible earlier. In addition, uL16m contains a

mitochondria-specific C-terminal extension that can promote its incorporation into the mtLSU even in the absence of a mature H89[40].

Besides its interaction with GTPBP7, GTPBP10 also contacts the biogenesis factor MALSU1 in intermediate 1. MALSU1 is a homolog of bacterial RsfS, which was proposed to cooperate with bacterial ObgE

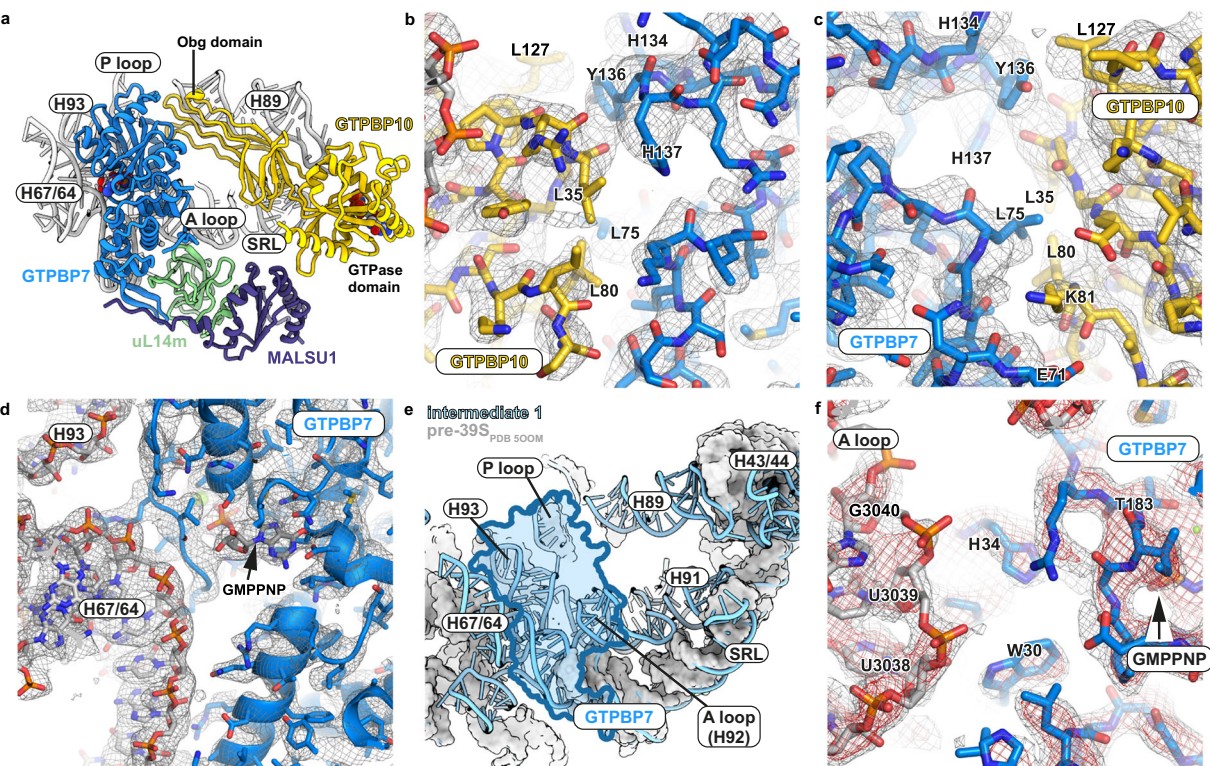

**Fig. 4 | Interactions of GTPBP7 with GTPBP10 and ribosomal RNA. a** Overview of GTPBP7 and GTPBP10 in the context of adjacent mitoribosomal RNA elements. SRL = sarcin-ricin loop (**b, c**) Two views of the interaction interface between GTPBP10 and GTPBP7 are shown with the experimental EM density. Interacting residues are highlighted. The sharpened, supersampled EM density is shown at 2.3 σ. **d** Interface of GTPBP7 and ribosomal RNA with the corresponding experimental EM density at 2.3 σ. **e** Overlay of the rRNA of intermediate 1 (light blue, cartoon) and an earlier maturation intermediate (gray, surface, PDB 5OOM)[7]. GTPBP7 is shown as silhouette at its binding site in intermediate 1 (blue). rRNA elements adjacent to GTPBP7 are not visible in the earlier maturation intermediate. **f** The location of histidine 34 (H34) of GTPBP7 in relation to the ribosomal A loop is depicted. EM density is shown at 2 different thresholds (2.3 σ (gray), 3.2 σ (red)).

during H89 maturation[39]. Previous studies on GTPBP5 concluded that, distinct to the bacterial system, it acts in concert with the N-terminal tail of biogenesis factors NSUN4 instead of MALSU1[9]. Our structural data now indicate that MALSU1 likely plays a role for GTPBP10 action as it engages in similar interactions with the GTPase as the bacterial RsfS (Fig. 3e–g, Supplementary Fig. 5d) and we do not find the N-terminal tail of NSUN4 to be close to GTPBP10. The interaction with MALSU1 could aid to position the nearby catalytic Walker B motif in switch II of the GTPBP10 GTPase domain on the SRL (Fig. 3e).

Overall, our data rationalize how sequential action of GTPBP10 and GTPBP5 replaces the function of bacterial ObgE in mitochondrial ribosome biogenesis. Our results display that the timing of important steps in mitochondrial ribosome biogenesis has been reorganized due to alterations in the involved biogenesis factors but also due to alterations in the ribosomal structure.

### Interactions of GTPBP7 with GTPBP10 and the ribosome

Our data suggest that GTPBP10 action requires the simultaneous presence of GTPBP7 on the ribosome for two reasons.

On the one hand, it stabilizes GTPBP10 and thereby also H89 on the ribosomal subunit interface (Fig. 4a, Supplementary Fig. 7a). While the tip of H89 is stabilized via interaction of GTPBP10 with the SRL and H91, the Obg domain stacks onto GTPBP7. The interfaces of both proteins show high shape complementarity and establish the interaction mostly via hydrophobic side chains including L35, L80, and L127 in GTPBP10 and L75, Y136, H137 in GTPBP7 (Fig. 4b, c). Moreover, we find a prominent density for the side chain of K81 from the Obg domain of GTPBP10 interacting with an alpha-helical turn formed by residues E71-L73 of GTPBP7 although our resolution does not allow to define the exact atomic interactions (Fig. 4c).

On the other hand, GTPBP7 may serve to stabilize adjacent rRNA elements to form a binding pocket for the rRNA H89. In intermediate 1, GTPBP7 makes extensive contact to ribosomal RNA elements including the A loop (H92), H93, and H64/67 (Fig. 4d, e). Most of these elements including H91-93 are largely invisible in earlier maturation intermediates but surround H89 in the mature mtLSU[7,10,40]. In accordance, the bacterial homolog RbgA was shown to stabilize related rRNA helices in A and P sites but not H89 in an immature 50S assembly intermediate in *B. subtilis* (Supplementary Fig. 7b, c)[35]. It is therefore tempting to speculate that GTPBP7 binding may be required to prepare a binding pocket in the ribosomal RNA to aid H89 accommodation via GTPBP10.

GTPBP7 was proposed to monitor methylation status of U3039, G3040, and G2815 in A and P loops, respectively[12]. Theoretically, intermediate 1 should at least carry a methylation at G3040 as the responsible methyltransferase MRM3 appears to act upstream of GTPBP10[41]. Due to the flexibility of the A loop tip, the resolution does however not permit to verify that G3040 is indeed modified in intermediate 1. In our reconstruction, GTPBP7 is too far from the P loop for a direct interaction but touches the backbone of the A loop (Fig. 4f). Our density does not provide an indication for stacking of A loop bases U3039 or G3040 with the catalytic residue H34 of GTPBP7 as proposed earlier[12]. H34 rather faces towards the gamma-phosphate of the bound nucleotide. 2'O-methyl groups of modified A loop nucleotides may instead directly interact with amino acid stretch P32/G33/H34 of GTPBP7 in our maturation intermediate although the local resolution does not permit to unambiguously identify atomic details of this putative interaction (Fig. 4f).

We also find GTPBP7 to bind with a beta-hairpin (residues 280-295) in a cleft between ribosomal protein uL14m and bL19m

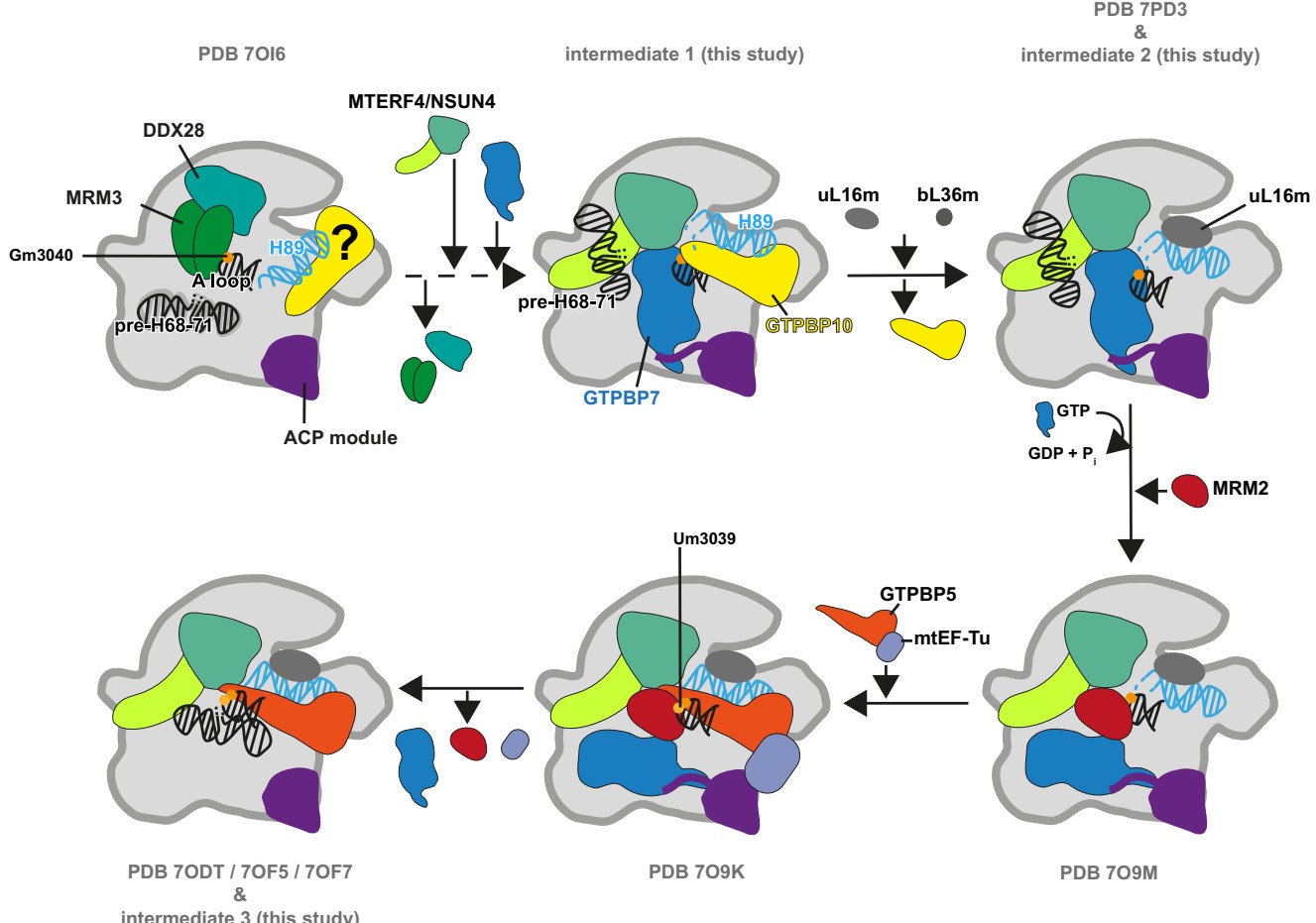

**Fig. 5 | Model for H89 maturation.** H89 maturation requires the concerted action of GTPBP10 and GTPBP5. GTPBP10 has earlier been proposed to be part of a maturation intermediate in conjunction with the methyltransferase MRM3 and helicase DDX28. However, the low resolution of the reconstruction renders this interpretation ambiguous. We now find GTPBP10 in concert with GTPBP7 bound to the maturing ribosome when the pre-H68-71 rRNA stretch has already been deposited on the NSUN4-MTERF4 dimer (intermediate 1). Localization of H89 into its ribosomal cavity enables proper incorporation of uL16m and bL36m, which in turn induce GTP hydrolysis in GTPBP10 and its dissociation from the ribosome. GTPBP7 may then remain on the mtLSU that contains a partially unstructured H89 base (intermediate 2, Supplementary Fig. 10). uL16m incorporation may finally trigger GTP hydrolysis and reorganization of GTPBP7 on the RNA surface. This will vacate the necessary space for binding of the methyltransferase MRM2, which installs the methylation on U3039 in the A loop. As shown earlier, binding of GTPBP5 completes folding of the H89 base (intermediate 3), releases the A loop from MRM2, and may trigger departure of MRM2 and GTPBP7 from the complex.

(Supplementary Fig. 7d). The GTPBP7 beta hairpin is additionally stabilized by the C-terminus of MALSU1 reaching over. The beta hairpin has earlier been shown to serve as an anchor point around which GTPBP7 swings from a conformation contacting the ribosomal immature PTC towards pre-RNA helices H68-71 upon binding of the methyltransferase MRM2 to the A loop[11,42].

Our reconstruction of GTPBP10 together with GTPBP7 now allows to rationalize previously described mutations reported to ablate GTPBP10 function in mitoribosome biogenesis[14]. We find the mutations to potentially either disturb the fold of the Obg domain and to hamper interaction with GTPBP7 (G82E), or to influence the interaction of GTPBP10 with 16S rRNA (deletion of R64 and K65) (Supplementary Fig. 8)[14].

## Discussion

We isolated mitoribosomal complexes from actively growing HEK293-6E cells in the presence of the non-hydrolysable nucleotide analog GMPPNP. Among the isolated complexes, we discovered a native biogenesis intermediate of the human mitoribosomal large subunit. Our intermediate contains 7 biogenesis factors including the heterodimer of NSUN4 and MTERF4, the MALSU1−L0R8F8−mtACP complex, and the two GTPases GTPBP7 and GTPBP10.

GTPBP10 is next to GTPBP5 one of two mitochondrial homologs of the bacterial GTPase ObgE. In bacteria, ObgE coordinates the folding of rRNA helix H89 and the incorporation of ribosomal proteins uL16 and bL36 to form a fully functional catalytic center[39]. Mitochondrial ribosome biogenesis requires two distinct homologs of the bacterial GTPase to generate functional ribosomes. Together with previously published structures of the methyltransferase MRM2 and GTPBP5, our data substantiate that the two mitochondrial homologs exert complementary but distinct functions in H89 folding. The structural data also allow us to derive a more complete picture of the order of events in mtLSU biogenesis (Fig. 5)[8,9,11].

Assembly of the PTC requires coordinated action of methyltransferases and GTPases to modify and fold the ribosomal RNA. In an earlier step, the methyltransferase MRM3 catalyzes the 2'-O-methylation of G3040 in the ribosomal A loop. After MRM3 is released, the NSUN4-MTERF4 dimer associates and leads to a rearrangement of the pre-H68-71 rRNA stretch. Then, GTPBP7 binding may stabilize RNA elements H91-93 that surround the H89 binding pocket on the mtLSU. GTPBP10 catalyzes in the presence of GTPBP7 the deposition of the already largely folded H89 into its ribosomal cavity while the base of the rRNA helix remains partially unfolded (intermediate 1). Upon H89 deposition, rearrangement of the L7/12

stalk base may lead to GTP hydrolysis in GTPBP10 eventually causing its dissociation from the ribosomal particle (intermediate 2, Supplementary Fig. 10). This assumption is strengthened by the observation that our reconstruction contains density that can account for an unhydrolysed guanosine triphosphate while the GTPBP5 intermediate with a stalk base conformation similar to the factor-free mtLSU was modeled with GDP. Our assignment of the nucleotide states of GTPBP7 and GTPBP10 has been deduced from comparison of our EM density with experimental structures of homologous ribosome biogenesis factors and their bound nucleotides from the eukaryotic cytosol and bacteria. Although our density analysis indicates that it is most probable that both GTPases have a guanosin triphosphate bound, the local resolution of our EM maps does not permit to completely rule out that GDP may be bound to either or both of the GTPases instead. Our current working model how GTPBP10 and GTPBP7 GTP hydrolysis is regulated and impacts the association of both factors with the maturing ribosome will therefore require further biochemical validation in future studies. In the bacterial ribosome, maturation of H89 has also been connected to motions in the stalk base that may ultimately lead to GTP hydrolysis and ObgE release although the exact conformational changes appear to be somewhat different (Supplementary Fig. 9d–i). H89 motion now allows rigid incorporation of uL16m that associates early to the rRNA and later on of bL36m. uL16m positioning may trigger GTP hydrolysis in GTPBP7 in analogy to the bacterial system, where incorporation of uL16 was shown to directly or indirectly stimulate GTPase activity of the bacterial GTPBP7 homolog RbgA[43]. Whether modification of A loop nucleotide G3040 also plays a role in activation of GTPBP7 at this stage remains to be clarified as the local density does not permit to confirm the absence or presence of the A loop RNA modifications in intermediate 1. GTP hydrolysis could then lead to a reorganization of GTPBP7 on the subunit interface around its hinge point associated to uL14m and bL19m. Now, the methyltransferase MRM2 can bind close to the PTC to catalyze 2'-O-methylation of U3039 in the A loop[11]. Finally, GTPBP5 binds to the maturation intermediate and triggers release of the A loop from the binding pocket in MRM2 and folding of the unstructured H89 base into its final conformation to complete formation of the PTC (intermediate 3)[8,9,11].

This division of tasks between GTPBP10 and GTPBP5 can be explained by structural features. GTPBP10 and GTPBP5 show largely similar folds but differ in their loop 1 and 3 regions as well as by peptide insertions in their Obg and GTPase domains. GTPBP5 contains longer loops 1 and 3, which contact immature rRNA elements of the catalytic center. Here, they coordinate the repositioning of the P loop and the folding of the base of H89 in conjunction with the N-terminal tail of NSUN4. Moreover, GTPBP5 was found to interact with the translation elongation factor mtEF-Tu, where mtEF-Tu was proposed to aid accommodation of GTPBP5 onto the mtLSU[11]. In contrast, GTPBP10 contains shorter loops 1 and 3 of the Obg domain and an alpha helical insertion that permit tight interactions of GTPBP10 with GTPBP7 and H89, respectively. The GTPBP10 loops do not primarily engage in protein-nucleic acid interaction at the PTC and the N-terminal tail of NSUN4 is disordered in our intermediate 1. Instead, the loops have rather been repurposed to form an interaction interface with GTPBP7 to stabilize GTPBP10 binding on the rRNA. One side of the Obg domain is enriched in positively charged amino acids enabling the tight interaction with the RNA backbone of H89 and its positioning on top of its binding cavity in the mtLSU. We do not find mtEF-Tu in our intermediate 1 assembly suggesting that GTPBP10 may be able to bind to the maturing ribosomal subunit by itself, possibly due to the additional contact to GTPBP7. Collectively, the structural data explain why GTPBP10 and GTPBP5 are both essential in mitoribosome biogenesis as only their concerted action completes H89 folding.

## Methods

### Cell culture

The human embryonic kidney 293EBNA1-6E cell line (HEK 293EBNA1-6E) was adapted to growth in suspension in F17 medium (FreeStyle™ F17 Expression Medium, Gibco), supplemented with 4 mM L-Glutamine (Sigma), 0.1% Pluronic F-68 (Gibco), and 1% heat-inactivated fetal bovine serum (Sigma). The cell culture was maintained around $1.5 \times 10^6$ cells/ml in 1L square polycarbonate storage bottles (Corning), and shaken in a humidified incubator (Infors HT) at 150 rpm and 37 °C with 5% $CO_2$. Cell density and viability were determined via the trypan blue exclusion method with the Countess™ automated cell counter (Invitrogen).

Polyethylenimine (PEI) was used as a transfection reagent for transient overexpression of mtRF1-AAG-3xFLAG in HEK 293EBNA1-6E cells, following a previously described method with some modifications[44]. Cell culture with viability >98% was split 3 h before transfection in fresh complete F17 medium to $1.0 \times 10^6$ cells/ml. Plasmid DNA of mtRF1-AAG-3xFLAG in a pcDNA3.1 vector backbone was ordered from Thermo Scientific. DNA and PEI at a ratio of 1:3 (w/w) were diluted in F17 medium to make up a transfection reagent volume as 1% of the culture volume, with final DNA concentration as 1 mg per liter of culture. The transfection mixture was incubated 20 min at room temperature to form polyplexes before being added to the culture. Two days after transfection, cells were harvested and used for the next steps.

### Mitochondria isolation

1L of HEK293-EBNA1-6E cell culture at a density of $2.5–3.0 \times 10^6$ cells/mL was harvested by centrifugation ($750 \times g$, 4 °C, 15 min) using a Sorvall SLC-4000 rotor (Thermo Fisher). Cells were washed in 20 mL of chilled phosphate-buffered saline (1X PBS, pH 7.4), and centrifuged ($1500 \times g$, 4 °C, 15 min) using the Centrifuge 5810R (Eppendorf). The cell pellet was subsequently resuspended in 15 mL of ice-cold RSB hypo buffer (10 mM NaCl, 1.5 mM $MgCl_2$, 10 mM Tris-HCl, pH 7.5) to allow cells to swell. After a 10 min incubation, swollen cells were broken by a Dounce homogenizer 100 mL tube and a B (tight) pestle with 15 strokes. The Dounce homogenizer was chilled on ice beforehand, and filled with a volume of 2.5X MS homogenization buffer (525 mM mannitol, 175 mM sucrose, 2.5 mM EDTA, 2.5 mM DTT, 12.5 mM Tris-HCl, pH 7.5) that was calculated accordingly to obtain a final concentration of 1X MS homogenization buffer after adding the cell suspension.

Mitochondria were isolated from the above homogenate by differential centrifugation. The supernatant containing mitoplasts was carefully collected after each round of centrifugation at $1300 \times g$ then $3000 \times g$ (4 °C, 15 min, Centrifuge 5810R). Crude mitochondria were pelleted by centrifugation at $9500 \times g$ (4 °C, 15 min) using the Optima XE-90 Ultracentrifuge with a Ti-45 rotor (Beckman Coulter). The final pellet was resuspended in 2 mL of Resuspension buffer (250 mM sucrose, 1 mM EDTA, 20 mM HEPES-KOH, pH 7.6), snap-frozen in liquid nitrogen and stored at −80 °C.

### Mitoribosome isolation

1 mL of mitochondria suspension (corresponding to 0.5 L HEK293-EBNA1-6E suspension cell culture at $1.5 − 2.0 \times 10^6$ cells/mL) were supplemented with 100 µL of 100 mM GMPPNP (Jena Bioscience, NU-899-50) and quickly thawn in a water bath at room temperature. The mitochondrial suspension was mixed with 1.75 mL of lysis buffer (20 mM HEPES-KOH pH 7.6, 100 mM KCl, 40 mM $MgCl_2$, 40 U/mL Ribolock RNase inhibitor (Thermo Scientific, 11581505), 1 tablet / 50 mL of Pierce Protease Inhibitors without EDTA (Thermo Scientific, 15677308), 0.8 mM spermidine pH 7.5, 1 mM DTT). 750 µL of 6x solubilization buffer (20 mM HEPES-KOH pH 7.6, 100 mM KCl, 40 mM $MgCl_2$, 9.6% (v/v) Triton X-100 (VWR, M143-1L), 1 mM DTT) were added and the sample was gently mixed by inversion. The final volume was around 3.5 mL in a 15 mL Falcon tube. The suspension was placed on

the rotation wheel for 20 min in the cold room for gentle agitation. Then the suspension was cleared 2 times for 10 min at 21300 × g and 4 °C. The supernatant was transferred into a fresh 15 mL Falcon tube. 1 open-top, thinwall, ultraclear ultracentrifugation tube (Beckman Coulter, 344057) was filled with 1.3 mL of 40% sucrose solution (20 mM HEPES-KOH pH 7.6, 100 mM KCl, 40 mM MgCl₂, 40% (w/v) sucrose, 1 mM DTT). 3.2 mL of the cleared, mitochondrial lysate was then carefully placed onto the sucrose cushion (ration of cushion:lysate was 1:2.5). The sample was spun for 4 h at 245,639 × g at 4 °C in a SW55 rotor using a Beckman Coulter Optima XE-90 ultracentrifuge. The supernatant was taken off and the pellets was rinsed 2 times with 500 µL of 1x monosome buffer (20 mM HEPES-KOH pH 7.6, 100 mM KCl, 40 mM MgCl₂, 1 mM DTT). The rinse was discarded and the pellets were submerged in 100 µL of 1x monosome buffer containing 1 mM GMPPNP. The pellets were resuspended by gentle shaking in the cold room for ~1 h. The remaining pellet was gently resuspended with the pipette and the ribosomal suspension was transferred into a 1.5 mL tube. The suspension was cleared 2 times at 21300 rcf for 10 min at 4 °C. The supernatant was tested in negative stain to verify the ribosome concentration, which was estimated to be around 70 nM. The suspension was used directly for cryo-grid preparation.

## Cryo-EM sample preparation and data collection

Quantifoil R 1.2/1.3 or R 2/2 plus C2 on 300 copper mesh were glow discharged for 30 s at 5 mA in a Leica Coater ACE 200. 5 µL of sample were applied to the grids and the sample was vitrified using a FEI Vitrobot Mark IV at 100% humidity, blot force 0, wait time of 30 s, and blotting times between 3 and 7 s. Two data sets were collected in linear mode as movies at a pixel size of 1.08 Å/px, 300 kV and 40 e/Å² with 33 or 42 fractions, respectively, on a FEI Titan Krios G2 equipped with Falcon 3 DED using EPU–Automated Acquisition Software (v2.14.0).

## Cryo-EM data analysis

Movies were aligned, dose weighted, and summed into micrographs using patch motion correction and patch CTF in CryoSPARC v.4.2.0[45]. Particles were identified using the blob picker tool with a minimum and maximum particle diameter of 150 Å and 350 Å, respectively. Afterwards picks were further filtered adjusting NCC score, as well as lower and upper local power thresholds to minimize false positive picks.

The classification scheme for intermediate 1 was as follows (Supplementary Fig. 11).

For data set 1, 8,901,726 particles were extracted from 35,399 micrographs with a box size of 480 px and Fourier cropped to 120 px. Particle images underwent 2D classification in batches for at least 60 online EM iterations. Well-resolved 2D classes were 2D classified one more time to remove remaining poor particle images from the image pool. The poor particle classes from the first 2D classification also underwent one more round of 2D classification to retrieve any good remaining particle images. A total of 2,764,332 particle images were finally selected for further 3D hetero refinement in 3 batches using reference volumes including 39S mtLSU volumes, 55S mitoribosome volumes, 28S mtSSU volumes, and 80S cytosolic ribosome volumes. Reference volumes were obtained via subset selection of 2D classes resembling the respective particle type and ab initio reconstruction with 2 classes. The ab initio class that was better resolved was then used as the reference volume.

At this point, it became obvious that 39S volumes showed partially inhomogeneous subunit interfaces hinting at the presence of 39S maturation intermediates in our ribosome preparation. To obtain these putative 39S maturation intermediates, the particle image subsets belonging to well-defined 39S 3D volumes were pooled, subjected to 3D homogeneous refinement, and further classified via local 3D classification in 10 classes without resolution restriction and without image alignment using a mask covering the 39S subunit interface (shown as blue semitransparent surface in the classification scheme). Classes that contained clear densities for maturation factors were pooled yielding 125,003 particle images. Classes where the subunit interface of the LSU did not show interpretable, defined density of immature rRNA and additional protein factors, or where the subunit interface was clearly in a mature state, were excluded from further classifications. Particle images were re-extracted at full pixel size and a box size of 480 px. They were local 3D classified into 4 classes with a mask surrounding the GTPase binding site between GAC and PTC. 1 class represented the mtLSU containing GTPBP7 and GTPBP10, one class contained only GTPBP5, and two classes contained only GTPBP7. The class containing GTPBP7 and GTPBP10 with 29,510 particle images was homogeneously refined and further local 3D classified using the same mask as in the step before to clean the particle population from false positive images that did not contain GTPBP10 yielding 17,274 good particle images.

For data set 2, 8,005,604 particle images were extracted from 30,287 micrographs with a box size of 480 px and Fourier cropped to 120 px. Particle images underwent the same 2D and initial 3D classification approaches as for dataset 1. However, after a clean 39S particle population was isolated, 558,320 particle images were reextracted at full pixel size of 1.08 Å with a box size of 480 px. They were homogenously refined and underwent local 3D classification using a mask covering the crevice between GAC and PTC (shown as semitransparent surface in the classification scheme) with a resolution cutoff of 10 Å and without image alignment. This classification yielded 10 classes, of which one class containing 50139 particle images showed density for GTPBP7 and density for a GTPase bound to the SRL, which we later identified to be GTPBP10. However, the occupancy did not appear to be high. To increase the occupancy, these particle images were homogenously refined and further local 3D classified. We tried various masks but eventually, we got the best separation of particle images using a mask that covered the entire mtLSU interface. The classification was done with a resolution cutoff of 6 Å and without alignment of particle images into 3 classes. One class of 14,382 particle images contained density for GTPBP7 and the other GTPase and the images were pooled with the 17,274 particle images obtained for the same complex from dataset 1.

The joined pool from dataset 1 and dataset 2 underwent homogenous refinement, local CTF estimation, and a final round of homogenous refinement resulting in a 3D reconstruction of 3.03 Å from 31,656 particle images.

The classification scheme for intermediate 2 was as follows (Supplementary Fig. 12).

Classification of particle images from dataset 1 was identical to the procedure for intermediate 1. After the second local 3D classification, however, the two classes with 68,335 particle images that contained GTPBP7 only were pooled with 89,161 particle images from dataset 2.

Initial processing for dataset 2 was identical to dataset 1. Once a clean 39S particle population was isolated, the particle images were reextracted at full pixel size of 1.08 Å with a box size of 480 px. They were homogenously refined and underwent local 3D classification using a mask covering almost the entire mtLSU subunit interface (shown as semitransparent surface in the classification scheme) with a resolution cutoff of 10 Å and without image alignment. Note that the mask was different from the one used in the intermediate 1 classification scheme. This classification yielded 10 classes, of which one class containing 89'161 particle images showed clear density for GTPBP7. These particle images were then joined with the particle images obtained from dataset 1.

The joined particle population from dataset 1 and dataset 2 was further cleaned via local 3D classification using a mask surrounding the crevice between GAC and PTC into three classes with a resolution cutoff of 6 Å without image alignment. The class that displayed the most detailed GTPBP7 EM density was used for final 3D homogenous

refinement yielding a reconstruction of 3.00 Å from 68,901 particle images.

The classification scheme for intermediate 3 was as follows (Supplementary Fig. 13).

Processing of datasets 1 and 2 followed the same strategy as outlined for intermediate 2. For dataset 1, 21,147 particle images were isolated after the second local 3D classification. They contained clear density for GTPBP5 and a more mature rRNA and were pooled with 33,129 particle images from dataset 2 that showed the same features.

The joined population of 60,273 particle images from dataset 1 and dataset 2 was further cleaned via local 3D classification using a mask surrounding the crevice between GAC and PTC into two classes using a resolution cutoff of 6 Å without image alignment. The class that displayed the most detailed GTPBP5 EM density was used for final 3D homogenous refinement yielding a reconstruction of 3.05 Å from 39,296 particle images.

Finally, to ensure that our classifications yielded distinct particle populations for intermediates 1–3, we have compared the particle images contributing to each reconstruction using the CryoSPARC 'Particle sets tool' and find that there is only very minor or no overlap between the particle image pools (232 particle images between intermediate 1 and 3, 357 particle images between intermediate 1 and 2, 0 particle images between intermediates 2 and 3).

## Model building

We decided to build structural models for intermediate 1 as it has not been described before, and for intermediate 2 as the currently published model contains a number of imperfections with respect to the connectivity of the rRNA backbone, the visibility of the rRNA surrounding the PTC, and the GTPBP7 active site.

For intermediate 1, we used the published LSU maturation intermediate containing GTPBP5 (PDB: 7ODT)[8] and AlphaFold models of GTPBP10 (referring to UniProt ID A4D1E9) and GTPBP7 (referring to UniProt ID Q9BT17) as starting point for model building. RNA elements that were not visible in our EM map were removed in Coot[46,47]. The Obg and GTPase domains of GTPBP10 were fitted separately as rigid bodies in UCSF Chimera v.1.15[48]. Afterwards, we revised the model manually in Coot (v.0.9.6). The RNA double helix of H89 was fitted as rigid body into its respective density and manually adjusted to account for changes especially in the tip and base region. All maturation factors as well as all ribosomal protein and ribosomal RNA were adjusted to account for the EM density either via manual model refinement or rigid body fitting of secondary structure elements and domains depending on the resolution of the respective region of the EM map. Non-hydrolysable nucleotide analog GMPPNP was added to GTPBP7 and GTPBP10 and coordinated with a Mg2+ ion and the respective residues in the active site of both GTPases. Due to limited resolutions in these areas, the side chains of uL16m, of the ribosomal stalk proteins, and of the GTPase domain of GTPBP10 were stripped to alanine. The final model was real space-refined in PHENIX v.1.19.2-4158 using default restraints (Ramachandran, C-beta deviations, rotamer, secondary structure) and global minimization as well as B factor refinement in 5 cycles with a weight of experimental data and restraints set to 1.5[49].

For intermediate 2, we used PDBs 7ODT, 7ODR, 7PD3, and the intermediate 1 model as starting points for model building in Coot (v.0.9.6)[8,12]. RNA elements from the different models were fused and manually remodeled where necessary to yield a model of the 16S rRNA that matched our experimental density. GTPBP7 was transplanted from our intermediate 1 model and slightly manually adjusted where necessary. Ribosomal proteins were chain-refined and, if necessary, manually adjusted. Finally, the model was real-space refined in PHENIX v.1.19.2-4158 using global minimization as well as B factor refinement in 5 cycles with a weight of experimental data and restraints set to 1.5[49]. Default restraints (Ramachandran, C-beta deviations, rotamer, secondary structure) were applied in combination with RNA model

restraints generated by DoubleHelix[50]. We used *checkMySequence* to judge the confidence of the fit of ribosomal RNA fragments in regions which showed ambiguous EM density[51].

Local resolution estimations have been carried out in cryoSPARC v.4.2.0. They are shown together with the angular distribution of particle images for intermediate 1 (Supplementary Fig. 14a), intermediate 2 (Supplementary Fig. 14b), and intermediate 3 (Supplementary Fig. 14c). FSCs of the half sets of the experimental data for intermediates 1, 2, and 3 as well as of the full map with the structural models for intermediate 1 and intermediate 2 have been calculated in PHENIX v.1.19.2-4158 using *phenix.mtriage* (Supplementary Fig. 15)[52]. Additionally, B factors have been plotted onto the structural model for intermediate 1 and intermediate 2 (Supplementary Fig. 15).

EM data collection and model validation parameters are provided in Supplementary Table 1.

## Figure preparation

Molecular graphics were generated using the PyMOL (Schroedinger), UCSF Chimera or ChimeraX packages[48,53].

## Reporting summary

Further information on research design is available in the Nature Portfolio Reporting Summary linked to this article.

## Data availability

Electron microscopy data have been deposited in the EMDB under accession codes EMD-17719 (intermediate 1), EMD-17720 (intermediate 2), and EMD-17721 (intermediate 3). The structural models for intermediate 1 and intermediate 2 have been deposited in the PDB database under the accession codes 8PK0 and 8QSJ, respectively. Raw movies from cryo-EM analysis are available from the corresponding author upon request.

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

## Acknowledgements

We would like to thank the staff at the Core Facility for Integrated Microscopy, Faculty of Health and Medical Sciences, University of Copenhagen for their support during EM data collection. Moreover, we would like to acknowledge Søren Kirk Amstrup for initial efforts in the model building process. This work has been supported by a Hallas-Møller Emerging Investigator grant from the Novo Nordisk Foundation to E. Kummer (NNF21OC0067360).

## Author contributions

E.K. designed the study and experiments. T.G.N. carried out the cell culture work and isolated mitochondria. T.G.N. and E.K. isolated mitoribosomes and vitrified samples. E.K. collected and analyzed EM data. E.K., T.G.N., and C.R. built and interpreted the structural models. E.K., T.G.N., and C.R. wrote the manuscript and prepared the figures.

## Competing interests

The authors declare no competing interests.
