## [Peer Review File · Nature Communications]

Structural insights into the role of GTPBP10 in the RNA maturation of the mitoribosomeREVIEWER COMMENTS

Reviewer #1 (Remarks to the Author):

This manuscript by Nguyen & Kummer describes the discovery of a novel assembly intermediate of the human mitochondrial large subunit which expands our knowledge of this pathway by connecting previously discovered intermediates. The authors computationally isolated this intermediate from a preparation of mitoribosomes that includes free 28S-like, free 39S-like, and 55S particles. The authors also report reconstructions of later assembly intermediates that have been previously solved from the same samples. The structure includes previously observed assembly factors (MALSU1-LOR8F8-mtACP, GTPBP7/MTG1, MTERF4-NSUN4, and GTPBP10) but in a novel complex. The conformation of rRNA (especially H68-71, H89, H43/44, H91, SRL, A loop) suggest the reconstruction represents a GTPBP10-bound state after MRM3/DDX28 dissociation. Interestingly, the authors show how human GTPases GTPBP10 and GTPBP5 replace the function of bacterial ObgE with distinct but related functions carried out by each of the GTPases. The authors propose a model for the folding of H89 and the peptidyl-transferase center which involves sequential binding of GTPBP10 then GTPBP5, along with binding of mitoribosomal proteins uL16m and bL36m.

The figures for the most part are clear and portray what is described in the text, and the manuscript is written clearly. This manuscript is suitable for publication in Nature Communications provided that all of the points listed below are addressed.

Major points:

1. Nucleotide states:

The authors have used GMPPNP throughout sample preparation and it seems like this fact was used to assign nucleotide states in both GTPBP10 and GTPBP7 sites as GMPPNP. However, a closer inspection of the provided cryo-EM map clearly shows that in both GTPB7 and GTPB10 the density around the nucleotide is ambiguous as the local resolution is not sufficient to distinguish between for example GDP or GMPPNP. For example, in the context of GTPB7 a GDP molecule can be fitted equally well if not better than GMPPNP. Since the presented cryo-EM data does not support any particular nucleotide state, the authors should clearly show the density for nucleotide binding sites for each GTPase. Separately, in the text they should address how the nucleotide state affects their model. Since the nucleotide state is very important to understand the mechanism of mitoribosomal assembly and since nucleotide hydrolysis can in principle be used to facilitate the incorporation of a GTPase or alternatively enable its release, the authors should outline the limitations of their study in this context as well.

2. Density maps for ligands:

As shown in Fig. S4D and S8A, the density for GMPPNP is ambiguous near GTPB10. As the local resolution should be used to guide the level to which the model is built and interpreted, the authors should amend their model to only contain what is clearly visible. In cases where a ligand or a chemical modification is built, the corresponding density map should be shown in a supplementary figure to show that there is sufficient detail in the maps. This relates to chemical modifications of pre-rRNA, nucleotide states of the GTPases and other ligands. For example, the currently presented map does not support the statement that G3040 is methylated (shown in Fig. 3C) so the model should be updated and reflect only what is observed.

3. Local resolution and model building:

The current model for intermediate 1 does not fully represent the experimental data as full side-chain containing models are present even in areas where the local resolution is closer to 5-6 Å. The authors should trim side chains in areas where these are clearly not visible. In parallel the authors should present their model colored by B-factors to highlight the degree of confidence with which models can be placed in different areas of the map.

4. Comparison with bacterial LSU assembly:

The authors' mechanistic interpretation of the data is that the addition of GMPPNP arrests GTPBP10 and GTPBP7 in a pre-hydrolysis, GTP-bound like state. The manuscript also draws comparisons between their structure of GTPBP10 to the bacterial homologue ObgE, which has been observed previously both in a native context (PDB 7BL2, 7BL3, 7BL5) and in a reconstituted context with a non-hydrolysable GTP analog (7BL6 and 7BL4). Here the authors should more thoroughly compare their structure (especially overall orientation of Obg and GTPase domains) of GTPBP10 and the surrounding area to these other structures rather than just 7BL2.

For example, the authors note that there is density, although weak, present for uL16m. However, the cryo-EM map clearly shows docking of uL16m, which should be shown in a figure. The weak density and low local resolution in the area could also suggest that there is still heterogeneity in these particles and that the data could either represent a local flexibility in uL16m or represent a mix of states. Performing local refinement and classification could improve resolution or improve how to interpret this weak uL16m density, although the authors have presumably tried this already. Separately, the authors should also comment on why in bacteria, ObgE is present in a GDP bound state after binding of uL16 (PDB 7BL5) and how this is compatible with their model.

5. Availability of all experimental data and clarity of cryo-EM processing:

The origin and processing scheme of cryo-EM data should be clarified and the authors should state this upfront. Firstly, the methods section describes that cells transiently expressing a mutant mtRF1 construct were used. This suggests that the primary target of this investigation was not mitoribosome assembly

and that the discovery of the described mitoribosomal assembly intermediates was a byproduct of unrelated studies. While this is fine, the authors should state this explicitly in the text. The authors should additionally upload their raw micrographs to EMPIAR to improve access for the entire community. Supplementary figures depicting the data processing scheme should be clearer. What criteria were used to exclude particular classes and can individual classes be annotated more? What proportion of particles are found in each class? For dataset 2, it appears that different volumes are shown for the initial classification of 39S particles for intermediate 1 vs. intermediates 2 and 3, and it seems impossible to find the reconstruction representing particles of intermediates 2 and 3 in the scheme for intermediate 1.

6. Placement of intermediate 2 in a mitoribosome assembly pathway:

The authors should cite and interpret their results in the context of a recent mitoribosome assembly review, which suggests that intermediate 2 (and PDB 7PD3) comes after MRM2 mediated A loop methylation, rather than prior, which is the model presented in this manuscript. A detailed analysis of experimental cryo-EM density of rRNA modifications and nucleotides in GTPases is important to clarify this discrepancy.

Minor points:

On lines 68, 143/144, & 265, the authors say “near-atomic EM reconstruction”. This phrase is unnecessary.

Fig. 4 should include NSUN4/MTERF4 coming in and DDX28/MRM3 leaving between PDB 7OI6 and intermediate 1 (this study). A dotted line or equivalent could be used to suggest multiple possible maturation steps.

In Fig. 3D, there is a label “aa 240-247” and it is unclear what that is referring to without reading the accompanying text. Additionally, the text describing this suggests that the GTPBP10 insertion clashes with uL11m, but this is not clear from the figure and both of these elements are present together here.

The description of ObgE as a “perfect chimera” (line 215) of both mitochondrial versions could be misleading to readers, perhaps just “chimera” as perfect suggests identical sequences.

Reviewer #2 (Remarks to the Author):

The manuscript by Ngyuen and Kummer reports the results of a structural investigation of the maturation of the human mitoribosome large subunit (mtLSU), at a late stage where most of the mitoribosomal proteins are associated and few rRNA modification, folding and anti-association factors still bound. The authors show three late-stage maturation intermediates of the mtLSU containing NSUN4-MTERF4 dimer along with the MALSU1-LOR8F8-mtACP trimer either along with GTPBP7 and 10, GTPBP7 only or GTPBP5 GTPases. Because the other two intermediates were thoroughly investigated by previous studies, the authors focus on the unprecedentedly observed intermediate containing GTPBP7 and 10.

The manuscript is well written and deals with a topic that will be of high interest to the communities of mRNA translation, (mito)ribosome maturation, structural biology and cryo-EM.

The figures are clear and all are necessary. However, the references are mainly focused on the human mitoribosome and neglect related publications from other species such as kinetoplastids and yeast.

I have a couple major comments and few minor ones, and I recommend the publication of the current manuscript provided that these issues are addressed:

Major points:

The authors speak of a "near-atomic resolution structure", yet on panel B of Figure2, from glancing the displayed map one can't see side chains or details consistent with a such level of resolution. While it's possible that the authors didn't pick the best view to demonstrate the resolution of their map, it is always of good practice to show the raw map at the area of interest or show a close-up on the segmentation of that area of interest so that the reader could appreciate the local resolution of the most important features of the reported structures. The same holds true for panels B and C of Figure3.

Is it possible that the authors' most significant map (GTPBP 7+10) is actually a mix of two independent classes of the late-stage maturing mtLSU with GTPBP7 or with GTPBP10? How could the authors ascertain the concomitant binding of these two different factors at the same time when the local resolution at this specific region can't allow accurate interpretation of map, as side chains are not distinguishable?

Minor points:

The authors must cite the work on the kinetoplastids' maturing mtLSU showing the involvement of GTPBP7 (RbgA).

Line 51: The authors present their structure as THE missing late-stage maturation intermediate, please rephrase, this structure is probably not only missing one!

Line 68: What does "near-atomic" refer to? The resolution? I would really advise against the use of such inaccurate terms, suffice it to communicate the resolution.

Line 144: The authors assign one of the densities as being GTPBP10. Did the authors consider validating their assignment by MS/MS?

Line 149: The authors mention a previous structure showing GTPBP10 in a different conformation and they hypothesize that this could be a preliminary conformation before complete accommodation of the latter. While this hypothesis might be true, is it possible that this previous structure was simply misinterpreted? I encourage the authors to attempt answering this question by simply fitting their model in this previously reported low-resolution structure.

REVIEWER COMMENTS

Reviewer #1 (Remarks to the Author):

This manuscript by Nguyen & Kummer describes the discovery of a novel assembly intermediate of the human mitochondrial large subunit which expands our knowledge of this pathway by connecting previously discovered intermediates. The authors computationally isolated this intermediate from a preparation of mitoribosomes that includes free 28S-like, free 39S-like, and 55S particles. The authors also report reconstructions of later assembly intermediates that have been previously solved from the same samples. The structure includes previously observed assembly factors (MALSU1-LOR8F8-mtACP, GTPBP7/MTG1, MTERF4-NSUN4, and GTPBP10) but in a novel complex. The conformation of rRNA (especially H68-71, H89, H43/44, H91, SRL, A loop) suggest the reconstruction represents a GTPBP10-bound state after MRM3/DDX28 dissociation. Interestingly, the authors show how human GTPases GTPBP10 and GTPBP5 replace the function of bacterial ObgE with distinct but related functions carried out by each of the GTPases. The authors propose a model for the folding of H89 and the peptidyl-transferase center which involves sequential binding of GTPBP10 then GTPBP5, along with binding of mitoribosomal proteins uL16m and bL36m.

The figures for the most part are clear and portray what is described in the text, and the manuscript is written clearly. This manuscript is suitable for publication in Nature Communications provided that all of the points listed below are addressed.

Major points:

1. Nucleotide states:

The authors have used GMPPNP throughout sample preparation and it seems like this fact was used to assign nucleotide states in both GTPBP10 and GTPBP7 sites as GMPPNP. However, a closer inspection of the provided cryo-EM map clearly shows that in both GTPB7 and GTPB10 the density around the nucleotide is ambiguous as the local resolution is not sufficient to distinguish between for example GDP or GMPPNP. For example, in the context of GTPB7 a GDP molecule can be fitted equally well if not better than GMPPNP. Since the presented cryo-EM data does not support any particular nucleotide state, the authors should clearly show the density for nucleotide binding sites for each GTPase. Separately, in the text they should address how the nucleotide state affects their model. Since the nucleotide state is very important to understand the mechanism of mitoribosomal assembly and since nucleotide hydrolysis can in principle be used to facilitate the incorporation of a GTPase or alternatively enable its release, the authors should outline the limitations of their study in this context as well.

ANSWER: We have included density maps for the nucleotide ligands in both GTPases to the Supplementary Figures (Supplementary Fig. 2).

In case of GTPBP7, we are confident to claim that our density represents rather GMPPNP than GDP. We infer this from the positioning of the Mg ion with respect to the phosphates, which albeit not at atomic resolution is still clearly visible in our density map and coordinated between Ser157 and Thr183 of GTPBP7 and the gamma and beta phosphates of GMPPNP. To be more certain, we have also overlaid our structure with recent structures of the eukaryotic homolog Nog2, which was visualized in the GTP and GDP bound state at high enough resolution to model the active site including potassium and magnesium ions (PDB 7UOO and PDB 7UQZ from Sekulski et al. 2023). From these overlays, it also appears more likely that our state represents a GMPPNP-bound state.

In case of GTPBP10, the nucleotide state is more ambivalent due to the lower resolution of the GTPase domain. However, we find that the GMPPNP we modelled into the density shows the better fit in comparison to a GDP molecule (Supplementary Fig. 2). For comparison of the GDP state, we superimposed the structural models of ObgE bound to GDP or GMPPNP from PDB 7BL5 and 7BL4,

respectively (Nikolay et al. 2021) with our model. Finally, we decided to keep GMPPNP in our structural model.

2. Density maps for ligands:

As shown in Fig. S4D and S8A, the density for GMPPNP is ambiguous near GTPB10. As the local resolution should be used to guide the level to which the model is built and interpreted, the authors should amend their model to only contain what is clearly visible. In cases where a ligand or a chemical modification is built, the corresponding density map should be shown in a supplementary figure to show that there is sufficient detail in the maps. This relates to chemical modifications of pre-rRNA, nucleotide states of the GTPases and other ligands. For example, the currently presented map does not support the statement that G3040 is methylated (shown in Fig. 3C) so the model should be updated and reflect only what is observed.

ANSWER: We believe that two things guide the model building process. The local resolution and prior knowledge of the structure. For example, it has been biochemically proven that methylation of G3040 is essential for proper mtLSU assembly and is carried out by MRM3. The current understanding is that MRM3 acts prior to H89 maturation and it is therefore conceivable that G3040 should be methylated in our maturation intermediate. That is why we had included it in the model although the tip of the A loop is relatively flexible in our reconstruction and the resolution is not sufficient to visualize the modification here. Nonetheless, we have now removed the RNA modification and show density maps for the ligands in NSUN4 (SAM) and mtACP (PM8) (Supplementary Fig. 1a and b) and the nucleotides bound to the active sites of GTPBP7 and GTPBP10 (Supplementary Fig. 2).

3. Local resolution and model building:

The current model for intermediate 1 does not fully represent the experimental data as full side-chain containing models are present even in areas where the local resolution is closer to 5-6 Å. The authors should trim side chains in areas where these are clearly not visible. In parallel the authors should present their model colored by B-factors to highlight the degree of confidence with which models can be placed in different areas of the map.

ANSWER: We have corrected our structural model according to the reviewers suggestions as follows: The side chains of the GTPBP10 GTPase domain have mostly been trimmed to alanine except for the region where GTPBP10 interacts with MALSU1 since clear side chain density is visible in this area (residues 190-198). Moreover, the L7/12 stalk base shows a relatively low resolution and has therefore been fitted as a rigid body into the experimental density during structure building. We have now in addition trimmed the side chains of the proteins in the stalk base and also removed the side chains from uL16m, which is only flexibly bound in intermediate 1.

Figures of the models of intermediate 1 and 2 coloured according to B factors have been added to Supplementary Fig 15.

4. Comparison with bacterial LSU assembly:

The authors' mechanistic interpretation of the data is that the addition of GMPPNP arrests GTPBP10 and GTPBP7 in a pre-hydrolysis, GTP-bound like state. The manuscript also draws comparisons between their structure of GTPBP10 to the bacterial homologue ObgE, which has been observed previously both in a native context (PDB 7BL2, 7BL3, 7BL5) and in a reconstituted context with a non-hydrolysable GTP analog (7BL6 and 7BL4). Here the authors should more thoroughly compare their structure (especially overall orientation of Obg and GTPase domains) of GTPBP10 and the surrounding area to these other structures rather than just 7BL2.

ANSWER: We have compared our structure to the various models from Nikolay et al. and find that the conformation of H89 is in most parts closest to state 1 (7BL2), which is why we have used it for the

comparison. In addition, the tip of H89 is differently structured in our intermediate than in any of the intermediates reported in Nikolay et al. Moreover, the Obg domain of GTPBP10 is in a distinct orientation than in the bacterial intermediates probably due to the difference in its loop structures, the interaction with GTPBP7, and because it engages with the RNA backbone of H89 more intimately than ObgE. We have added overlay figures with 7BL2, 7BL5, and 7BL4 now for comparison (Supplementary Fig. 9).

Although the L7/12 stalk base is rather flexible in our intermediate, we are able to fit the stalk base as a rigid body accounting very well for a lowpass-filtered version of our density (Supplementary Fig. 5e). In our model, we find the stalk base to be rotated outwards in comparison to the mature mitoribosome and the bacterial maturation intermediates (Fig. 2 f-h, Supplementary Fig. 9d-i). This is mostly due to the mitoribosomal-specific, extended conformation of the H89 tip, which pushes against H43/H44 at the stalk base (Fig. 2c and h). Consequently, H89 deposition and folding of its tip into the final conformation is required for the stalk base to swing back into its more inward facing position, which may be part of the mechanism to induce nucleotide hydrolysis in GTPBP10.

In contrast to our structure, bacterial intermediates 7BL6 and 7BL4 already contain bL36, which means that they are later maturation intermediates and a comparison is therefore difficult. In fact, a direct comparison between the bacterial and mitochondrial maturation intermediates is in general not straightforward. One reason is that the bacterial ribosome contains RNA elements that are absent or shortened in the mitochondrial case, for example H38. In the bacterial structures 7BL2, 7BL3, and 7BL5, H89 positioning occurs together with changes in H38 and final binding of uL16 is only observed in state 3 (7BL5) (Suppl. Fig. 4 a-c). The reason is that H38 essentially blocks access to the uL16 binding site as long as H89 is not in an almost fully mature position. However, as H38 is substantially shorter in mitoribosomes, the uL16m binding site appears to be accessible earlier in the maturation process, which is probably why we can already see uL16m density in our reconstruction despite H89 being far from its final mature position (Suppl. Fig. 4e and Suppl. Fig. 5f).

Another reason that complicates a direct comparison of the bacterial and mitochondrial systems is the fact that while bacteria contain only one Obg protein, mitochondria work with the consecutive action of two proteins that carry out distinct tasks as evidenced in this manuscript. This also implies that the temporal organization of H89 maturation is likely distinct in mitochondria because we do not assume that bacterial ObgE binds to the maturing particle twice but that it finalizes its tasks within one round of binding. Eventually, it is also evident that the overall structure of ObgE is more similar to GTPBP5 than GTPBP10 in terms of Obg domain and relative positioning of the Obg domain on the ribosomal interface and with respect to H89, which also makes it more difficult to assume that GTPBP10 acts according to the exact same principles as ObgE (Suppl. Fig. 4a-e).

A direct comparison between bacterial ObgE and GTPBP10 therefore comes with a number of uncertainties whose discussion we believe would occupy too much space in this manuscript. We have added now some more detail to the text and added additional information to supplementary figures as indicated above. However, we refrain from extensive comparisons to the bacterial system in the main manuscript due to the above-mentioned differences between both systems.

In addition to the requested comparison to bacterial ObgE, we have for completeness also included a comparison with RbgA – the bacterial equivalent of GTPBP7 (Suppl. Fig. 7b and c).

We have added following statement to the main text:

‘Moreover, while ribosomal protein uL16 joins the LSU in bacteria only when ObgE has almost completed H89 accommodation, it is present in mitoribosomes already when H89 is far from its final position (Supplementary Fig. 4). This is possible because the bacterial rRNA element H38 is significantly shortened in mitoribosomes and consequently the binding site for uL16m becomes accessible earlier.

In addition, uL16m contains a mitochondria-specific C-terminal extension that can promote its incorporation into the mtLSU even in the absence of a mature H89.⁴⁰

For example, the authors note that there is density, although weak, present for uL16m. However, the cryo-EM map clearly shows docking of uL16m, which should be shown in a figure.

ANSWER: We have added a figure that shows that uL16 has docked but appears to be still flexibly deposited as evidenced by the partial lack of density, for example for the region between residues 55 and 127, and the overall low resolution. (Fig. 2b, Supplementary Fig. 5f)

The weak density and low local resolution in the area could also suggest that there is still heterogeneity in these particles and that the data could either represent a local flexibility in uL16m or represent a mix of states. Performing local refinement and classification could improve resolution or improve how to interpret this weak uL16m density, although the authors have presumably tried this already.

Answer: We have done an additional focused classification for uL16 but do not find any discrete particle populations with and without uL16, which indicates that the low resolution is rather due to flexibility.

Separately, the authors should also comment on why in bacteria, ObgE is present in a GDP bound state after binding of uL16 (PDB 7BL5) and how this is compatible with their model.

ANSWER: We believe that Nikolay et al. propose a model, in which maturation of H89 by ObgE allows uL16 binding and leads to inward motion of the GAC and a repositioning of the GTPase domain at the GAC, which in turn triggers GTP hydrolysis in ObgE. We assume that this is why 7BL5 contains GDP instead of GTP although it appears that Nikolay et al. do not specifically state the reason for including GDP in 7BL5 in their manuscript. They derive their catalytic model from a comparison of ObgE bound to an immature particle with GDP and ObgE bound to a mature particle with GMPPNP.

For our model, we believe that H89 deposition by GTPBP10 will also lead to stable binding of uL16m (and eventually bL36m although not seen in our intermediate yet) and inward motion of the GAC. These conformational changes in the ribosomal particle are - as in the model of Nikolay et al. - likely causing repositioning of the GTPBP10 catalytic site on the SRL and could lead to ordering of the switch 2 region (which is disordered in our intermediate 1) to induce GTP hydrolysis in GTPBP10. In this regard, the models make similar assumptions, but we do think that the direct comparison between bacterial and mitochondrial systems is difficult due to the differences in mitoribosomal structure and the existence of two distinct ObgE homologs in mitochondria as detailed already above. Intriguingly, GTPBP5 in PDB 7ODT (Lenarcic et al. 2021) shows an inward positioned GAC and it has been proposed to contain GDP, while our intermediate has an outward oriented GAC and contains GMPPNP. Taken together, we believe that our assignment of the nucleotide state and the model we propose, are coherent with earlier published observations.

We have added following statement to the text:

‘Upon H89 deposition, rearrangement of the L7/12 stalk base may lead to GTP hydrolysis in GTPBP10 eventually causing its dissociation from the ribosomal particle. This assumption is strengthened by the observation that our reconstruction contains density that can account for an unhydrolyzed guanosine triphosphate while the GTPBP5 intermediate with a stalk base conformation similar to the factor-free mtLSU was modelled with GDP. In the bacterial ribosome, maturation of H89 has also been connected to motions in the stalk base that may ultimately lead to GTP hydrolysis and ObgE release although the exact conformational changes appear to be somewhat different (Supplementary Fig. 9d-i).’

5. Availability of all experimental data and clarity of cryo-EM processing:

The origin and processing scheme of cryo-EM data should be clarified and the authors should state this upfront.

Firstly, the methods section describes that cells transiently expressing a mutant mtRF1 construct were used. This suggests that the primary target of this investigation was not mitoribosome assembly and that the discovery of the described mitoribosomal assembly intermediates was a byproduct of unrelated studies. While this is fine, the authors should state this explicitly in the text.

ANSWER: We have now rephrased the text to:

‘We initially intended to trap mitoribosomal translation termination complexes with a catalytic inactive mutant of the termination factor mtRF1, which has been shown to decode the non-canonical stop codons AGG and AGA in human mitochondria.¹⁸⁻²⁰ To this end, we purified mitoribosomes in the presence of the nonhydrolyzable nucleotide analog GMPPNP from HEK293-6E cell, in which we had transiently overexpressed mtRF1-AAG-3xFLAG. The mitoribosomal pool was analyzed by single particle cryoEM. Although we did not identify a particle subset containing mtRF1, we computationally isolated the mitochondrial ribosome in complex with translation elongation factor mtEFG1, the initiating 55S mitoribosome containing mtIF2, a contamination of 80S cytoplasmic ribosomal particles with elongation factor eEF2 bound, as well as 3 distinct mtLSU maturation intermediates (Fig. 1a).’

The authors should additionally upload their raw micrographs to EMPIAR to improve access for the entire community.

ANSWER: We appreciate the suggestion to upload our data sets to EMPIAR and will consider it.

Supplementary figures depicting the data processing scheme should be clearer. What criteria were used to exclude particular classes and can individual classes be annotated more?

ANSWER: We have excluded classes where the subunit interface of the LSU did not show any indication for interpretable, defined density of immature rRNA and additional protein factors, or where the subunit interface was clearly in a mature state. We have tried various classification approaches including 3D variability analysis vs. 3D classification, various local masks, different numbers of 3D classes, various resolution cutoffs, different particle pools and others to extract maturation intermediates. The intermediates that we present in this manuscript were the ones we were able to obtain from the dataset and we could not annotate any other maturation states in the particles although there are additional classes where the RNA is clearly not yet mature. We however could not identify density for maturation factors in these classes, maybe because the involved biogenesis factors bind less stably and have dissociated during sample preparation.

While revising the figure, we realized that we had accidentally made a mistake in boxing the classes that were selected after the first 3D classification in dataset 1. We had boxed classes 4,5, and 6 instead of 5,6, and 7. We have now corrected the mistake.

What proportion of particles are found in each class?

ANSWER: We have added the class distributions in % to the classification overviews.

For dataset 2, it appears that different volumes are shown for the initial classification of 39S particles for intermediate 1 vs. intermediates 2 and 3, and it seems impossible to find the reconstruction representing particles of intermediates 2 and 3 in the scheme for intermediate 1.

ANSWER: We have expanded the description of the classifications schemes in the method section to make them more clear. We have used different masks for the first local 3D classification of 39S particles in dataset 2 for intermediate 1 in comparison to intermediates 2 and 3 (the mask is indicated by the blue, transparent volumes shown together with the input average reconstruction in the

classification scheme). This means that the particles assigned to each class are not the same for both classification schemes thereby leading to slightly different initial volumes. As mentioned above, we have used various approaches to yield the cleanest and highest resolution volumes for each state. This means that our emphasis was not to obtain reconstructions for all intermediates from one classification scheme but the best reconstruction for each intermediate with the most suitable classification scheme. Using different classification schemes was especially important in our case as our intermediates represent only a small subset of the entire dataset, which made it more challenging to extract them from the particle pool with sufficient purity and resolution. To make sure that we are looking at distinct particle populations in our reconstructions for intermediates 1-3, we have compared the particle images in each reconstruction using the CryoSPARC 'Particle sets tool' and find that there is only very minor or no overlap between the classes (232 particle images between intermediate 1 and 3, 357 particle images between intermediate 1 and 2, 0 particle images between intermediates 2 and 3).

There are classes that contain the maturation factors of intermediates 2 and 3 in the scheme for intermediate 1 but the small size of the volume images may make it unfortunately somewhat complicated to see them properly. For example, in the second 3D classification of dataset 1, the first class shows density for GTPBP5 (intermediate 3) and classes 2 and 4 contain density for GTPBP7 only (intermediate 2).

6. Placement of intermediate 2 in a mitoribosome assembly pathway:

The authors should cite and interpret their results in the context of a recent mitoribosome assembly review, which suggests that intermediate 2 (and PDB 7PD3) comes after MRM2 mediated A loop methylation, rather than prior, which is the model presented in this manuscript. A detailed analysis of experimental cryo-EM density of rRNA modifications and nucleotides in GTPases is important to clarify this discrepancy.

ANSWER: We are not entirely sure, which review article the reviewer refers to but assume it may be: Khawaja et al. Trends in Biochemical Sciences (2023). We think that the positioning of intermediate 2 is not correct in Khawaja et al. Instead, we think that it rather corresponds to a position suggested in Lavdovskaia et al., another review on mitoribosome biogenesis published in 2021. We think that Khawaja et al. are wrong because the RNA segment corresponding to H68-71 has clearly not been accommodated in intermediate 2. However, in intermediate 3, we find H71 already positioned on the ribosomal subunit interface. Although it may be possible, we don't consider it likely that GTPBP7 has the capacity to revert H71 binding, and to re-deposit it onto MTERF4. This argues against the model proposed by Khawaja et al, which positions intermediate 3 before intermediate 2.

We believe that one reason why Khawaja et al place intermediate 2 after intermediate 3 is based on the suggestion that GTPBP7 monitors the methylation status of U3039 by stacking its catalytic residue H34 onto the base, which has been suggested by Chandrasekaran et al. However, neither their EM density nor ours provides solid evidence that H34 of GTPBP7 indeed stacks onto U3039. We are moreover not aware of any conclusive evidence in the literature that would proof that the role of GTPBP7 is indeed to monitor methylation of the A and P loops in mitochondria but this assumption has been extrapolated from other ribosomal systems, for example the cytoplasmic GTPBP7 equivalent Nog2. In addition, this misinterpretation may also be based on the fact that Chandrasekaran et al. have built the base of H89 in a mature conformation although their density (as well as ours) does not support this assumption as the modelled regions are not well resolved indicating that they are still flexible and immature (Supplementary Fig. 10). This would argue that intermediate 2 needs to be placed in the maturation pathway before GTPBP5 action and consequently before MRM2 action because GTPBP5 is responsible for displacement of the A loop from the MRM2 catalytic center. This would indicate that GTPBP7 binds the ribosome before G3039 has been methylated and it is therefore unclear if it indeed monitors G3039 methylation status. Overall, we believe that our model is in light

of the experimental data, the more likely one and that GTPBP7 placement in the maturation pathway in Khawaja et al. has been misled by wrong interpretations in an earlier intermediate 2 model. Because of these misinterpretations and since the model from Chadrasekaran et al. contains a number of technical flaws such as frequent chain breaks in the ribosomal RNA and errors in the catalytic center of GTPBP7, we have decided to revise the model and deposit a new version of this biogenesis intermediate in the PDB database for the community.

Minor points:

On lines 68, 143/144, & 265, the authors say “near-atomic EM reconstruction”. This phrase is unnecessary.

ANSWER: We have removed ‘near-atomic’ throughout the manuscript.

Fig. 4 should include NSUN4/MTERF4 coming in and DDX28/MRM3 leaving between PDB 7O16 and intermediate 1 (this study). A dotted line or equivalent could be used to suggest multiple possible maturation steps.

ANSWER: We have updated the model accordingly.

In Fig. 3D, there is a label “aa 240-247” and it is unclear what that is referring to without reading the accompanying text. Additionally, the text describing this suggests that the GTPBP10 insertion clashes with uL11m, but this is not clear from the figure and both of these elements are present together here.

ANSWER: We have corrected the figure (now Fig. 2f)

The description of ObgE as a “perfect chimera” (line 215) of both mitochondrial versions could be misleading to readers, perhaps just “chimera” as perfect suggests identical sequences.

ANSWER: We have changed the sentence and removed ‘perfect’.

Reviewer #2 (Remarks to the Author):

The manuscript by Ngyuen and Kummer reports the results of a structural investigation of the maturation of the human mitoribosome large subunit (mtLSU), at a late stage where most of the mitoribosomal proteins are associated and few rRNA modification, folding and anti-association factors still bound. The authors show three late-stage maturation intermediates of the mtLSU containing NSUN4-MTERF4 dimer along with the MALSU1-LOR8F8-mtACP trimer either along with GTPBP7 and 10, GTPBP7 only or GTPBP5 GTPases. Because the other two intermediates were thoroughly investigated by previous studies, the authors focus on the unprecedentedly observed intermediate containing GTPBP7 and 10.

The manuscript is well written and deals with a topic that will be of high interest to the communities of mRNA translation, (mito)ribosome maturation, structural biology and cryo-EM.

The figures are clear and all are necessary. However, the references are mainly focused on the human mitoribosome and neglect related publications from other species such as kinetoplastids and yeast.

I have a couple major comments and few minor ones, and I recommend the publication of the current manuscript provided that these issues are addressed:

Major points:

The authors speak of a "near-atomic resolution structure", yet on panel B of Figure2, from glancing the displayed map one can't see side chains or details consistent with a such level of resolution. While it's possible that the authors didn't pick the best view to demonstrate the resolution of their map, it is always of good practice to show the raw map at the area of interest or show a close-up on the segmentation of that area of interest so that the reader could appreciate the local resolution of the most important features of the reported structures. The same holds true for panels B and C of Figure3.

ANSWER: The term 'near-atomic' refers to the overall resolution of the map but we understand that the local resolution varies and the term may therefore be misleading. We have consequently removed it throughout the manuscript. As the reviewer suggests, we have included more panels with the experimental EM density in main and supplementary figures to provide the reader with an impression of the depth of the local data.

Is it possible that the authors' most significant map (GTPBP 7+10) is actually a mix of two independent classes of the late-stage maturing mtLSU with GTPBP7 or with GTPBP10? How could the authors ascertain the concomitant binding of these two different factors at the same time when the local resolution at this specific region can't allow accurate interpretation of map, as side chains are not distinguishable?

ANSWER: We have analyzed the data in intermediate 1 by local classification with various masks including tight masks that encircle only GTPBP10 or GTPBP7. Local classification was restricted to a resolution of 6 Angstrom and is therefore not based on the visibility of side chains. We were not able to separate the intermediate 1 data into distinct populations that contain only GTPBP7 or GTPBP10-bound particles, arguing that both GTPases are bound at the same time. We also do not find traces for H89 density in the accommodated for, which we would expect to see if the reconstruction was just a mix of GTPBP10 bound particles with immature H89 and GTPBP7 bound particles with accommodated H89. Moreover, the interfaces of GTPBP7 and GTPBP10 are clearly very compatible in their shape and closely associated, which is for example in stark contrast to GTPBP5 for which we can also not identify any class where it is bound to the pre-39S together with GTPBP7. These observations indicate that intermediate 1 is a true maturation intermediate and not just an artificial mixture of various intermediates due to poor image separation.

In addition, we do find many side chains in the interface to be well visible, for example L35, L80, and K81 of GTPBP10, and Y136, or H137 in GTPBP7. We can therefore be certain about the accuracy of our model. We have included now additional images with our experimental EM density to clarify the level of detail at the interface (Fig. 4b and c) and have added additional information to the main text.

'The interfaces of both proteins show high shape complementarity and establish the interaction mostly via hydrophobic side chains including L35, L80, and L127 in GTPBP10 and L75, Y136, H137 in GTPBP7 (Fig. 4b and c). Moreover, we find a prominent density for the side chain of K81 from the Obg domain of GTPBP10 interacting with an alpha-helical turn formed by residues E71-L73 of GTPBP7 although our resolution does not allow to define the exact atomic interactions.'

Minor points:

The authors must cite the work on the kinetoplastids' maturing mtLSU showing the involvement of GTPBP7 (RbgA).

ANSWER: We thank the reviewer for this comment. We had included references to kinetoplastid maturation but did not specify it in the text and have now also realized that the reference list was not complete. We apologize for this mistake and have corrected it.
The text now reads:

‘GTPBP7 is a homolog of biogenesis factor RbgA, which facilitates incorporation of bL36 in bacteria.²⁴ It adopts a position on the maturing mtLSU earlier observed only in the absence of the methyltransferase MRM2 as well as in kinetoplastid mtLSU biogenesis.^{12,24-27}

Line 51: The authors present their structure as THE missing late-stage maturation intermediate, please rephrase, this structure is probably not only missing one!

ANSWER: We have rephrased to:

‘Despite the wealth of mtLSU maturation snapshots, additional key intermediates have so far escaped structural elucidation. Here, we present the structure of one missing late-stage maturation intermediate of the human mtLSU.’

Line 68: What does "near-atomic" refer to? The resolution? I would really advise against the use of such inaccurate terms, suffice it to communicate the resolution.

ANSWER: We have removed ‘near-atomic’ throughout the manuscript.

Line 144: The authors assign one of the densities as being GTPBP10. Did the authors consider validating their assignment by MS/MS?

ANSWER: We have initially considered it. However, the GTPBP10-containing intermediate represents only a small subset of the entire dataset and we believe that MS will therefore not be able to provide us with a reliable answer. Firstly, the peptide coverage will be presumably quite poor and therefore nonconclusive as we assume that we would identify various biogenesis factors in the MS data that we copurify with the ribosomal particles – even some that may not be clearly visible in the EM reconstructions. Secondly, we would still measure over the entire ensemble without an indication that the peptides we identify belong to intermediate 1 instead of just being a contamination carried through the purification or bound to junk particles that we dismissed from the classification. We are however confident that our assignment is correct as we can clearly identify the GTPase and Obg domains, which limits the number of possible candidates to GTPBP5 and GTPBP10 in mitochondria. From those, only the Obg domain of GTPBP10 can account for the density in our map as the loops in GTPBP5 would be too long to fit into the reconstruction. Moreover, we can clearly assign a number of side chains in the Obg domain of GTPBP10 strengthening our interpretation.

Line 149: The authors mention a previous structure showing GTPBP10 in a different conformation and they hypothesize that this could be a preliminary conformation before complete accommodation of the latter. While this hypothesis might be true, is it possible that this previous structure was simply misinterpreted? I encourage the authors to attempt answering this question by simply fitting their model in this previously reported low-resolution structure.

ANSWER: Our model of intermediate 1 does not fit the previously reported density. The density for GTPBP10 in the previous published reconstruction is extremely weak and of very poor resolution, which also does not allow us to fit our model of GTPBP10 separately into this density or to be sure that the density represents GTPBP10. While our interpretation is consistent with the orientation of Obg proteins previously shown in GTPBP5-containing intermediates and with the structural data from bacterial ObgE, as well as with our common biochemical understanding of how GTPases dock onto the

SRL, the structural model in PDB 7OI6 is more difficult to rationalize. It is therefore possible that the previous structure could be a misinterpretation. We have rephrased the text accordingly.

from:

'A low resolution cryo-EM structure has earlier identified GTPBP10 at the maturing mtLSU interface, but has placed it in a non-canonical conformation with the Obg instead of the GTPase domain contacting the SRL.¹⁰ Also, the rRNA helix H89 is in a distinct conformation in that reconstruction and displays major clashes with the putative GTPBP10.'

to:

'A low resolution cryo-EM structure has earlier claimed to have identified GTPBP10 at the maturing mtLSU interface, but has placed it in a non-canonical conformation with the Obg instead of the GTPase domain contacting the SRL.¹⁰ Also, the rRNA helix H89 is shown in a distinct conformation in that model and displays major clashes with the putative GTPBP10. Overall, the resolution of the previous reconstruction can be considered too low to assign the density with certainty to GTPBP10.'

We have also deleted following sentence from the manuscript.

'It could be possible that the earlier report shows a pre-accommodation conformation of GTPBP10 whereas our structure displays it in the final accommodated, catalytically active position.'

REVIEWERS' COMMENTS

Reviewer #1 (Remarks to the Author):

In the revised version of this manuscript, Nguyen & Kummer have provided additional supplementary data. However, it is the opinion of this reviewer, that the authors have not been able to sufficiently address the prior concerns so that publication in a more specialized journal may be preferable. There are a number of scientific reasons for this, some of which are listed below.

First, and most importantly, the additional supplementary panels (c.f. Supplementary figure 2) that should support the notion that ligands can be identified from the experimental data to deduce a molecular mechanism, do not accomplish this task. At the current resolution, the authors simply cannot assign nucleotide states. Structural models are representations of what can be directly observed in the maps, and it is in the opinion of this reviewer that this standard is not met. Superimposing structures from another species and a completely different particle that was refined into a different map, will not be enough to explain what nucleotide is present in the current particle if the underlying experimental data does not have sufficient resolution to enable a precise assignment. The absence of sufficient resolution in the maps will therefore require biochemical experiments to decipher the mechanism of the described GTPases, which the authors may want to consider for a future submission.

Second, uncertainty of experimental data is not a problem per se, but should be fully disclosed, be it at the level of resolution or B-factors. The authors seem to have used a limited range to map both B-factors and resolutions of their maps (c.f. Supplementary figures 14 & 15), which give the readers the impression that for example B-factors shown in supplementary figure 15 peak at 200 for intermediate 1 whereas the model of intermediate 1 provided for the first submission had B-factors ranging from 0 to more than 450.

Third, the notion that a large dataset is collected for multiple manuscripts but that it may be okay for these data to be withheld from the scientific community seems incompatible with the spirit of open science at Nature Communications. The response to a prior comment ("The authors should additionally upload their raw micrographs to EMPIAR to improve access for the entire scientific community"), which is: "We appreciate the suggestion to upload our data sets to EMPIAR and will consider it", does not suggest that the authors see that this is a real problem under the circumstances. Given the nature of the data and the fraction of the data that is part of this manuscript, publication of this manuscript in any journal should absolutely require prior acceptance of the raw data at EMPIAR to provide the scientific community with access to all of the underlying data.

In summary, the data as presented in the current manuscript are not sufficient to support the claims of the authors.

Reviewer #2 (Remarks to the Author):

The authors have addressed appropriately the raised concerns.

REVIEWER COMMENTS

REVISION 1

Reviewer #1 (Remarks to the Author):

This manuscript by Nguyen & Kummer describes the discovery of a novel assembly intermediate of the human mitochondrial large subunit which expands our knowledge of this pathway by connecting previously discovered intermediates. The authors computationally isolated this intermediate from a preparation of mitoribosomes that includes free 28S-like, free 39S-like, and 55S particles. The authors also report reconstructions of later assembly intermediates that have been previously solved from the same samples. The structure includes previously observed assembly factors (MALSU1-LOR8F8-mtACP, GTPBP7/MTG1, MTERF4-NSUN4, and GTPBP10) but in a novel complex. The conformation of rRNA (especially H68-71, H89, H43/44, H91, SRL, A loop) suggest the reconstruction represents a GTPBP10-bound state after MRM3/DDX28 dissociation. Interestingly, the authors show how human GTPases GTPBP10 and GTPBP5 replace the function of bacterial ObgE with distinct but related functions carried out by each of the GTPases. The authors propose a model for the folding of H89 and the peptidyl-transferase center which involves sequential binding of GTPBP10 then GTPBP5, along with binding of mitoribosomal proteins uL16m and bL36m.

The figures for the most part are clear and portray what is described in the text, and the manuscript is written clearly. This manuscript is suitable for publication in Nature Communications provided that all of the points listed below are addressed.

Major points:

1. Nucleotide states:

The authors have used GMPPNP throughout sample preparation and it seems like this fact was used to assign nucleotide states in both GTPBP10 and GTPBP7 sites as GMPPNP. However, a closer inspection of the provided cryo-EM map clearly shows that in both GTPB7 and GTPB10 the density around the nucleotide is ambiguous as the local resolution is not sufficient to distinguish between for example GDP or GMPPNP. For example, in the context of GTPB7 a GDP molecule can be fitted equally well if not better than GMPPNP. Since the presented cryo-EM data does not support any particular nucleotide state, the authors should clearly show the density for nucleotide binding sites for each GTPase. Separately, in the text they should address how the nucleotide state affects their model. Since the nucleotide state is very important to understand the mechanism of mitoribosomal assembly and since nucleotide hydrolysis can in principle be used to facilitate the incorporation of a GTPase or alternatively enable its release, the authors should outline the limitations of their study in this context as well.

ANSWER: We have included density maps for the nucleotide ligands in both GTPases to the Supplementary Figures (Supplementary Fig. 2).

In case of GTPBP7, we are confident to claim that our density represents rather GMPPNP than GDP. We infer this from the positioning of the MG ion with respect to the phosphates, which albeit not at atomic resolution is still clearly visible in our density map and coordinated between Ser157 and Thr183 of GTPBP7 and the gamma and beta phosphates of GMPPNP. To be more certain, we have also overlaid our structure with recent structures of the eukaryotic homolog Nog2, which was visualized in the GTP and GDP bound state at high enough resolution to model the active site including potassium and magnesium ions (PDB 7UOO and PDB 7UQZ from Sekulski et al. 2023). From these overlays, it also appears more likely that our state represents a GMPPNP-bound state.

In case of GTPBP10, the nucleotide state is more ambivalent due to the lower resolution of the GTPase domain. However, we find that the GMPPNP we modelled into the density shows the better fit in

comparison to a GDP molecule (Supplementary Fig. 2). For comparison of the GDP state, we superimposed the structural models of ObgE bound to GDP or GMPPNP from PDB 7BL5 and 7BL4, respectively (Nikolay et al. 2021) with our model. Finally, we decided to keep GMPPNP in our structural model.

2. Density maps for ligands:

As shown in Fig. S4D and S8A, the density for GMPPNP is ambiguous near GTPB10. As the local resolution should be used to guide the level to which the model is built and interpreted, the authors should amend their model to only contain what is clearly visible. In cases where a ligand or a chemical modification is built, the corresponding density map should be shown in a supplementary figure to show that there is sufficient detail in the maps. This relates to chemical modifications of pre-rRNA, nucleotide states of the GTPases and other ligands. For example, the currently presented map does not support the statement that G3040 is methylated (shown in Fig. 3C) so the model should be updated and reflect only what is observed.

ANSWER: We believe that two things guide the model building process. The local resolution and prior knowledge of the structure. For example, it has been biochemically proven that methylation of G3040 is essential for proper mtLSU assembly and is carried out by MRM3. The current understanding is that MRM3 acts prior to H89 maturation and it is therefore conceivable that G3040 should be methylated in our maturation intermediate. That is why we had included it in the model although the tip of the A loop is relatively flexible in our reconstruction and the resolution is not sufficient to visualize the modification here. Nonetheless, we have now removed the RNA modification and show density maps for the ligands in NSUN4 (SAM) and mtACP (PM8) (Supplementary Fig. 1a and b) and the nucleotides bound to the active sites of GTPBP7 and GTPBP10 (Supplementary Fig. 2).

3. Local resolution and model building:

The current model for intermediate 1 does not fully represent the experimental data as full side-chain containing models are present even in areas where the local resolution is closer to 5-6 Å. The authors should trim side chains in areas where these are clearly not visible. In parallel the authors should present their model colored by B-factors to highlight the degree of confidence with which models can be placed in different areas of the map.

ANSWER: We have corrected our structural model according to the reviewers suggestions as follows: The side chains of the GTPBP10 GTPase domain have mostly been trimmed to alanine except for the region where GTPBP10 interacts with MALSU1 since clear side chain density is visible in this area (residues 190-198). Moreover, the L7/12 stalk base shows a relatively low resolution and has therefore been fitted as a rigid body into the experimental density during structure building. We have now in addition trimmed the side chains of the proteins in the stalk base and also removed the side chains from uL16m, which is only flexibly bound in intermediate 1.

Figures of the models of intermediate 1 and 2 coloured according to B factors have been added to Supplementary Fig 15.

4. Comparison with bacterial LSU assembly:

The authors' mechanistic interpretation of the data is that the addition of GMPPNP arrests GTPBP10 and GTPBP7 in a pre-hydrolysis, GTP-bound like state. The manuscript also draws comparisons between their structure of GTPBP10 to the bacterial homologue ObgE, which has been observed previously both in a native context (PDB 7BL2, 7BL3, 7BL5) and in a reconstituted context with a non-hydrolysable GTP analog (7BL6 and 7BL4). Here the authors should more thoroughly compare their structure (especially overall orientation of Obg and GTPase domains) of GTPBP10 and the surrounding area to these other structures rather than just 7BL2.

ANSWER: We have compared our structure to the various models from Nikolay et al. and find that the conformation of H89 is in most parts closest to state 1 (7BL2), which is why we have used it for the comparison. In addition, the tip of H89 is differently structured in our intermediate than in any of the intermediates reported in Nikolay et al. Moreover, the Obg domain of GTPBP10 is in a distinct orientation than in the bacterial intermediates probably due to the difference in its loop structures, the interaction with GTPBP7, and because it engages with the RNA backbone of H89 more intimately than ObgE. We have added overlay figures with 7BL2, 7BL5, and 7BL4 now for comparison (Supplementary Fig. 9).

Although the L7/12 stalk base is rather flexible in our intermediate, we are able to fit the stalk base as a rigid body accounting very well for a lowpass-filtered version of our density (Supplementary Fig. 5e). In our model, we find the stalk base to be rotated outwards in comparison to the mature mitoribosome and the bacterial maturation intermediates (Fig. 2 f-h, Supplementary Fig. 9d-i). This is mostly due to the mitoribosomal-specific, extended conformation of the H89 tip, which pushes against H43/H44 at the stalk base (Fig. 2c and h). Consequently, H89 deposition and folding of its tip into the final conformation is required for the stalk base to swing back into its more inward facing position, which may be part of the mechanism to induce nucleotide hydrolysis in GTPBP10.

In contrast to our structure, bacterial intermediates 7BL6 and 7BL4 already contain bL36, which means that they are later maturation intermediates and a comparison is therefore difficult. In fact, a direct comparison between the bacterial and mitochondrial maturation intermediates is in general not straightforward. One reason is that the bacterial ribosome contains RNA elements that are absent or shortened in the mitochondrial case, for example H38. In the bacterial structures 7BL2, 7BL3, and 7BL5, H89 positioning occurs together with changes in H38 and final binding of uL16 is only observed in state 3 (7BL5) (Suppl. Fig. 4 a-c). The reason is that H38 essentially blocks access to the uL16 binding site as long as H89 is not in an almost fully mature position. However, as H38 is substantially shorter in mitoribosomes, the uL16m binding site appears to be accessible earlier in the maturation process, which is probably why we can already see uL16m density in our reconstruction despite H89 being far from its final mature position (Suppl. Fig. 4e and Suppl. Fig. 5f).

Another reason that complicates a direct comparison of the bacterial and mitochondrial systems is the fact that while bacteria contain only one Obg protein, mitochondria work with the consecutive action of two proteins that carry out distinct tasks as evidenced in this manuscript. This also implies that the temporal organization of H89 maturation is likely distinct in mitochondria because we do not assume that bacterial ObgE binds to the maturing particle twice but that it finalizes its tasks within one round of binding. Eventually, it is also evident that the overall structure of ObgE is more similar to GTPBP5 than GTPBP10 in terms of Obg domain and relative positioning of the Obg domain on the ribosomal interface and with respect to H89, which also makes it more difficult to assume that GTPBP10 acts according to the exact same principles as ObgE (Suppl. Fig. 4a-e).

A direct comparison between bacterial ObgE and GTPBP10 therefore comes with a number of uncertainties whose discussion we believe would occupy too much space in this manuscript. We have added now some more detail to the text and added additional information to supplementary figures as indicated above. However, we refrain from extensive comparisons to the bacterial system in the main manuscript due to the above-mentioned differences between both systems.

In addition to the requested comparison to bacterial ObgE, we have for completeness also included a comparison with RbgA – the bacterial equivalent of GTPBP7 (Suppl. Fig. 7b and c).

We have added following statement to the main text:

‘Moreover, while ribosomal protein uL16 joins the LSU in bacteria only when ObgE has almost completed H89 accommodation, it is present in mitoribosomes already when H89 is far from its final position (Supplementary Fig. 4). This is possible because the bacterial rRNA element H38 is significantly

shortened in mitoribosomes and consequently the binding site for uL16m becomes accessible earlier. In addition, uL16m contains a mitochondria-specific C-terminal extension that can promote its incorporation into the mtLSU even in the absence of a mature H89.⁴⁰

For example, the authors note that there is density, although weak, present for uL16m. However, the cryo-EM map clearly shows docking of uL16m, which should be shown in a figure.

ANSWER: We have added a figure that shows that uL16 has docked but appears to be still flexibly deposited as evidenced by the partial lack of density, for example for the region between residues 55 and 127, and the overall low resolution. (Fig. 2b, Supplementary Fig. 5f)

The weak density and low local resolution in the area could also suggest that there is still heterogeneity in these particles and that the data could either represent a local flexibility in uL16m or represent a mix of states. Performing local refinement and classification could improve resolution or improve how to interpret this weak uL16m density, although the authors have presumably tried this already.

Answer: We have done an additional focused classification for uL16 but do not find any discrete particle populations with and without uL16, which indicates that the low resolution is rather due to flexibility.

Separately, the authors should also comment on why in bacteria, ObgE is present in a GDP bound state after binding of uL16 (PDB 7BL5) and how this is compatible with their model.

ANSWER: We believe that Nikolay et al. propose a model, in which maturation of H89 by ObgE allows uL16 binding and leads to inward motion of the GAC and a repositioning of the GTPase domain at the GAC, which in turn triggers GTP hydrolysis in ObgE. We assume that this is why 7BL5 contains GDP instead of GTP although it appears that Nikolay et al. do not specifically state the reason for including GDP in 7BL5 in their manuscript. They derive their catalytic model from a comparison of ObgE bound to an immature particle with GDP and ObgE bound to a mature particle with GMPPNP.

For our model, we believe that H89 deposition by GTPBP10 will also lead to stable binding of uL16m (and eventually bL36m although not seen in our intermediate yet) and inward motion of the GAC. These conformational changes in the ribosomal particle are - as in the model of Nikolay et al. - likely causing repositioning of the GTPBP10 catalytic site on the SRL and could lead to ordering of the switch 2 region (which is disordered in our intermediate 1) to induce GTP hydrolysis in GTPBP10. In this regard, the models make similar assumptions, but we do think that the direct comparison between bacterial and mitochondrial systems is difficult due to the differences in mitoribosomal structure and the existence of two distinct ObgE homologs in mitochondria as detailed already above. Intriguingly, GTPBP5 in PDB 7ODT (Lenarcic et al. 2021) shows an inward positioned GAC and it has been proposed to contain GDP, while our intermediate has an outward oriented GAC and contains GMPPNP. Taken together, we believe that our assignment of the nucleotide state and the model we propose, are coherent with earlier published observations.

We have added following statement to the text:

'Upon H89 deposition, rearrangement of the L7/12 stalk base may lead to GTP hydrolysis in GTPBP10 eventually causing its dissociation from the ribosomal particle. This assumption is strengthened by the observation that our reconstruction contains density that can account for an unhydrolyzed guanosine triphosphate while the GTPBP5 intermediate with a stalk base conformation similar to the factor-free mtLSU was modelled with GDP. In the bacterial ribosome, maturation of H89 has also been connected to motions in the stalk base that may ultimately lead to GTP hydrolysis and ObgE release although the exact conformational changes appear to be somewhat different (Supplementary Fig. 9d-i).'

5. Availability of all experimental data and clarity of cryo-EM processing:

The origin and processing scheme of cryo-EM data should be clarified and the authors should state this upfront.

Firstly, the methods section describes that cells transiently expressing a mutant mtRF1 construct were used. This suggests that the primary target of this investigation was not mitoribosome assembly and that the discovery of the described mitoribosomal assembly intermediates was a byproduct of unrelated studies. While this is fine, the authors should state this explicitly in the text.

ANSWER: We have now rephrased the text to:

'We initially intended to trap mitoribosomal translation termination complexes with a catalytic inactive mutant of the termination factor mtRF1, which has been shown to decode the non-canonical stop codons AGG and AGA in human mitochondria.¹⁸⁻²⁰ To this end, we purified mitoribosomes in the presence of the nonhydrolyzable nucleotide analog GMPPNP from HEK293-6E cell, in which we had transiently overexpressed mtRF1-AAG-3xFLAG. The mitoribosomal pool was analyzed by single particle cryoEM. Although we did not identify a particle subset containing mtRF1, we computationally isolated the mitochondrial ribosome in complex with translation elongation factor mtEFG1, the initiating 55S mitoribosome containing mtIF2, a contamination of 80S cytoplasmic ribosomal particles with elongation factor eEF2 bound, as well as 3 distinct mtLSU maturation intermediates (Fig. 1a).'

The authors should additionally upload their raw micrographs to EMPIAR to improve access for the entire community.

ANSWER: We appreciate the suggestion to upload our data sets to EMPIAR and will consider it.

Supplementary figures depicting the data processing scheme should be clearer. What criteria were used to exclude particular classes and can individual classes be annotated more?

ANSWER: We have excluded classes where the subunit interface of the LSU did not show any indication for interpretable, defined density of immature rRNA and additional protein factors, or where the subunit interface was clearly in a mature state. We have tried various classification approaches including 3D variability analysis vs. 3D classification, various local masks, different numbers of 3D classes, various resolution cutoffs, different particle pools and others to extract maturation intermediates. The intermediates that we present in this manuscript were the ones we were able to obtain from the dataset and we could not annotate any other maturation states in the particles although there are additional classes where the RNA is clearly not yet mature. We however could not identify density for maturation factors in these classes, maybe because the involved biogenesis factors bind less stably and have dissociated during sample preparation.

While revising the figure, we realized that we had accidentally made a mistake in boxing the classes that were selected after the first 3D classification in dataset 1. We had boxed classes 4,5, and 6 instead of 5,6, and 7. We have now corrected the mistake.

What proportion of particles are found in each class?

ANSWER: We have added the class distributions in % to the classification overviews.

For dataset 2, it appears that different volumes are shown for the initial classification of 39S particles for intermediate 1 vs. intermediates 2 and 3, and it seems impossible to find the reconstruction representing particles of intermediates 2 and 3 in the scheme for intermediate 1.

ANSWER: We have expanded the description of the classifications schemes in the method section to make them more clear. We have used different masks for the first local 3D classification of 39S

particles in dataset 2 for intermediate 1 in comparison to intermediates 2 and 3 (the mask is indicated by the blue, transparent volumes shown together with the input average reconstruction in the classification scheme). This means that the particles assigned to each class are not the same for both classification schemes thereby leading to slightly different initial volumes. As mentioned above, we have used various approaches to yield the cleanest and highest resolution volumes for each state. This means that our emphasis was not to obtain reconstructions for all intermediates from one classification scheme but the best reconstruction for each intermediate with the most suitable classification scheme. Using different classification schemes was especially important in our case as our intermediates represent only a small subset of the entire dataset, which made it more challenging to extract them from the particle pool with sufficient purity and resolution. To make sure that we are looking at distinct particle populations in our reconstructions for intermediates 1-3, we have compared the particle images in each reconstruction using the CryoSPARC 'Particle sets tool' and find that there is only very minor or no overlap between the classes (232 particle images between intermediate 1 and 3, 357 particle images between intermediate 1 and 2, 0 particle images between intermediates 2 and 3).

There are classes that contain the maturation factors of intermediates 2 and 3 in the scheme for intermediate 1 but the small size of the volume images may make it unfortunately somewhat complicated to see them properly. For example, in the second 3D classification of dataset 1, the first class shows density for GTPBP5 (intermediate 3) and classes 2 and 4 contain density for GTPBP7 only (intermediate 2).

6. Placement of intermediate 2 in a mitoribosome assembly pathway:

The authors should cite and interpret their results in the context of a recent mitoribosome assembly review, which suggests that intermediate 2 (and PDB 7PD3) comes after MRM2 mediated A loop methylation, rather than prior, which is the model presented in this manuscript. A detailed analysis of experimental cryo-EM density of rRNA modifications and nucleotides in GTPases is important to clarify this discrepancy.

ANSWER: We are not entirely sure, which review article the reviewer refers to but assume it may be: Khawaja et al. Trends in Biochemical Sciences (2023). We think that the positioning of intermediate 2 is not correct in Khawaja et al. Instead, we think that it rather corresponds to a position suggested in Lavdovskaia et al., another review on mitoribosome biogenesis published in 2021. We think that Khawaja et al. are wrong because the RNA segment corresponding to H68-71 has clearly not been accommodated in intermediate 2. However, in intermediate 3, we find H71 already positioned on the ribosomal subunit interface. Although it may be possible, we don't consider it likely that GTPBP7 has the capacity to revert H71 binding, and to re-deposit it onto MTERF4. This argues against the model proposed by Khawaja et al, which positions intermediate 3 before intermediate 2.

We believe that one reason why Khawaja et al place intermediate 2 after intermediate 3 is based on the suggestion that GTPBP7 monitors the methylation status of U3039 by stacking its catalytic residue H34 onto the base, which has been suggested by Chandrasekaran et al. However, neither their EM density nor ours provides solid evidence that H34 of GTPBP7 indeed stacks onto U3039. We are moreover not aware of any conclusive evidence in the literature that would proof that the role of GTPBP7 is indeed to monitor methylation of the A and P loops in mitochondria but this assumption has been extrapolated from other ribosomal systems, for example the cytoplasmic GTPBP7 equivalent Nog2. In addition, this misinterpretation may also be based on the fact that Chandrasekaran et al. have built the base of H89 in a mature conformation although their density (as well as ours) does not support this assumption as the modelled regions are not well resolved indicating that they are still flexible and immature (Supplementary Fig. 10). This would argue that intermediate 2 needs to be placed in the maturation pathway before GTPBP5 action and consequently before MRM2 action because GTPBP5 is responsible for displacement of the A loop from the MRM2 catalytic center. This would indicate that GTPBP7 binds the ribosome before G3039 has been methylated and it is therefore

unclear if it indeed monitors G3039 methylation status. Overall, we believe that our model is in light of the experimental data, the more likely one and that GTPBP7 placement in the maturation pathway in Khawaja et al. has been misled by wrong interpretations in an earlier intermediate 2 model. Because of these misinterpretations and since the model from Chadrsekaran et al. contains a number of technical flaws such as frequent chain breaks in the ribosomal RNA and errors in the catalytic center of GTPBP7, we have decided to revise the model and deposit a new version of this biogenesis intermediate in the PDB database for the community.

Minor points:

On lines 68, 143/144, & 265, the authors say “near-atomic EM reconstruction”. This phrase is unnecessary.

ANSWER: We have removed ‘near-atomic’ throughout the manuscript.

Fig. 4 should include NSUN4/MTERF4 coming in and DDX28/MRM3 leaving between PDB 7O16 and intermediate 1 (this study). A dotted line or equivalent could be used to suggest multiple possible maturation steps.

ANSWER: We have updated the model accordingly.

In Fig. 3D, there is a label “aa 240-247” and it is unclear what that is referring to without reading the accompanying text. Additionally, the text describing this suggests that the GTPBP10 insertion clashes with uL11m, but this is not clear from the figure and both of these elements are present together here.

ANSWER: We have corrected the figure (now Fig. 2f)

The description of ObgE as a “perfect chimera” (line 215) of both mitochondrial versions could be misleading to readers, perhaps just “chimera” as perfect suggests identical sequences.

ANSWER: We have changed the sentence and removed ‘perfect’.

Reviewer #2 (Remarks to the Author):

The manuscript by Ngyuen and Kummer reports the results of a structural investigation of the maturation of the human mitoribosome large subunit (mtLSU), at a late stage where most of the mitoribosomal proteins are associated and few rRNA modification, folding and anti-association factors still bound. The authors show three late-stage maturation intermediates of the mtLSU containing NSUN4-MTERF4 dimer along with the MALSU1-LOR8F8-mtACP trimer either along with GTPBP7 and 10, GTPBP7 only or GTPBP5 GTPases. Because the other two intermediates were thoroughly investigated by previous studies, the authors focus on the unprecedentedly observed intermediate containing GTPBP7 and 10.

The manuscript is well written and deals with a topic that will be of high interest to the communities of mRNA translation, (mito)ribosome maturation, structural biology and cryo-EM.

The figures are clear and all are necessary. However, the references are mainly focused on the human mitoribosome and neglect related publications from other species such as kinetoplastids and yeast.

I have a couple major comments and few minor ones, and I recommend the publication of the current

manuscript provided that these issues are addressed:

Major points:

The authors speak of a "near-atomic resolution structure", yet on panel B of Figure 2, from glancing the displayed map one can't see side chains or details consistent with a such level of resolution. While it's possible that the authors didn't pick the best view to demonstrate the resolution of their map, it is always of good practice to show the raw map at the area of interest or show a close-up on the segmentation of that area of interest so that the reader could appreciate the local resolution of the most important features of the reported structures. The same holds true for panels B and C of Figure 3.

ANSWER: The term 'near-atomic' refers to the overall resolution of the map but we understand that the local resolution varies and the term may therefore be misleading. We have consequently removed it throughout the manuscript. As the reviewer suggests, we have included more panels with the experimental EM density in main and supplementary figures to provide the reader with an impression of the depth of the local data.

Is it possible that the authors' most significant map (GTPBP 7+10) is actually a mix of two independent classes of the late-stage maturing mtLSU with GTPBP7 or with GTPBP10? How could the authors ascertain the concomitant binding of these two different factors at the same time when the local resolution at this specific region can't allow accurate interpretation of map, as side chains are not distinguishable?

ANSWER: We have analyzed the data in intermediate 1 by local classification with various masks including tight masks that encircle only GTPBP10 or GTPBP7. Local classification was restricted to a resolution of 6 Angstrom and is therefore not based on the visibility of side chains. We were not able to separate the intermediate 1 data into distinct populations that contain only GTPBP7 or GTPBP10-bound particles, arguing that both GTPases are bound at the same time. We also do not find traces for H89 density in the accommodated for, which we would expect to see if the reconstruction was just a mix of GTPBP10 bound particles with immature H89 and GTPBP7 bound particles with accommodated H89. Moreover, the interfaces of GTPBP7 and GTPBP10 are clearly very compatible in their shape and closely associated, which is for example in stark contrast to GTPBP5 for which we can also not identify any class where it is bound to the pre-39S together with GTPBP7. These observations indicate that intermediate 1 is a true maturation intermediate and not just an artificial mixture of various intermediates due to poor image separation.

In addition, we do find many side chains in the interface to be well visible, for example L35, L80, and K81 of GTPBP10, and Y136, or H137 in GTPBP7. We can therefore be certain about the accuracy of our model. We have included now additional images with our experimental EM density to clarify the level of detail at the interface (Fig. 4b and c) and have added additional information to the main text.

'The interfaces of both proteins show high shape complementarity and establish the interaction mostly via hydrophobic side chains including L35, L80, and L127 in GTPBP10 and L75, Y136, H137 in GTPBP7 (Fig. 4b and c). Moreover, we find a prominent density for the side chain of K81 from the Obg domain of GTPBP10 interacting with an alpha-helical turn formed by residues E71-L73 of GTPBP7 although our resolution does not allow to define the exact atomic interactions.'

Minor points:

The authors must cite the work on the kinetoplastids' maturing mtLSU showing the involvement of GTPBP7 (RbgA).

ANSWER: We thank the reviewer for this comment. We had included references to kinetoplastid maturation but did not specify it in the text and have now also realized that the reference list was not complete. We apologize for this mistake and have corrected it.

The text now reads:

'GTPBP7 is a homolog of biogenesis factor RbgA, which facilitates incorporation of bL36 in bacteria.²⁴ It adopts a position on the maturing mtLSU earlier observed only in the absence of the methyltransferase MRM2 as well as in kinetoplastid mtLSU biogenesis.^{12,24-27}

Line 51: The authors present their structure as THE missing late-stage maturation intermediate, please rephrase, this structure is probably not only missing one!

ANSWER: We have rephrased to:

'Despite the wealth of mtLSU maturation snapshots, additional key intermediates have so far escaped structural elucidation. Here, we present the structure of one missing late-stage maturation intermediate of the human mtLSU.'

Line 68: What does "near-atomic refer to? The resolution? I would really advise against the use of such inaccurate terms, suffice it to communicate the resolution.

ANSWER: We have removed 'near-atomic' throughout the manuscript.

Line 144: The authors assign one of the densities as being GTPBP10. Did the authors consider validating their assignment by MS/MS?

ANSWER: We have initially considered it. However, the GTPBP10-containing intermediate represents only a small subset of the entire dataset and we believe that MS will therefore not be able to provide us with a reliable answer. Firstly, the peptide coverage will be presumably quite poor and therefore nonconclusive as we assume that we would identify various biogenesis factors in the MS data that we copurify with the ribosomal particles – even some that may not be clearly visible in the EM reconstructions. Secondly, we would still measure over the entire ensemble without an indication that the peptides we identify belong to intermediate 1 instead of just being a contamination carried through the purification or bound to junk particles that we dismissed from the classification.

We are however confident that our assignment is correct as we can clearly identify the GTPase and Obg domains, which limits the number of possible candidates to GTPBP5 and GTPBP10 in mitochondria. From those, only the Obg domain of GTPBP10 can account for the density in our map as the loops in GTPBP5 would be too long to fit into the reconstruction. Moreover, we can clearly assign a number of side chains in the Obg domain of GTPBP10 strengthening our interpretation.

Line 149: The authors mention a previous structure showing GTPBP10 in a different conformation and they hypothesize that this could be a preliminary conformation before complete accommodation of the latter. While this hypothesis might be true, is it possible that this previous structure was simply misinterpreted? I encourage the authors to attempt answering this question by simply fitting their model in this previously reported low-resolution structure.

ANSWER: Our model of intermediate 1 does not fit the previously reported density. The density for GTPBP10 in the previous published reconstruction is extremely weak and of very poor resolution, which also does not allow us to fit our model of GTPBP10 separately into this density or to be sure that the density represents GTPBP10. While our interpretation is consistent with the orientation of Obg proteins previously shown in GTPBP5-containing intermediates and with the structural data from bacterial ObgE, as well as with our common biochemical understanding of how GTPases dock onto the

SRL, the structural model in PDB 7OI6 is more difficult to rationalize. It is therefore possible that the previous structure could be a misinterpretation. We have rephrased the text accordingly.

from:

‘A low resolution cryo-EM structure has earlier identified GTPBP10 at the maturing mtLSU interface, but has placed it in a non-canonical conformation with the Obg instead of the GTPase domain contacting the SRL.¹⁰ Also, the rRNA helix H89 is in a distinct conformation in that reconstruction and displays major clashes with the putative GTPBP10.’

to:

‘A low resolution cryo-EM structure has earlier claimed to have identified GTPBP10 at the maturing mtLSU interface, but has placed it in a non-canonical conformation with the Obg instead of the GTPase domain contacting the SRL.¹⁰ Also, the rRNA helix H89 is shown in a distinct conformation in that model and displays major clashes with the putative GTPBP10. Overall, the resolution of the previous reconstruction can be considered too low to assign the density with certainty to GTPBP10.’

We have also deleted following sentence from the manuscript.

‘It could be possible that the earlier report shows a pre-accommodation conformation of GTPBP10 whereas our structure displays it in the final accommodated, catalytically active position.’

REVISION 2

Reviewer #1 (Remarks to the Author)

In the revised version of this manuscript, Nguyen & Kummer have provided additional supplementary data. However, it is the opinion of this reviewer, that the authors have not been able to sufficiently address the prior concerns so that publication in a more specialized journal may be preferable. There are a number of scientific reasons for this, some of which are listed below.

First, and most importantly, the additional supplementary panels (c.f. Supplementary figure 2) that should support the notion that ligands can be identified from the experimental data to deduce a molecular mechanism, do not accomplish this task. At the current resolution, the authors simply cannot assign nucleotide states. Structural models are representations of what can be directly observed in the maps, and it is in the opinion of this reviewer that this standard is not met. Superimposing structures from another species and a completely different particle that was refined into a different map, will not be enough to explain what nucleotide is present in the current particle if the underlying experimental data does not have sufficient resolution to enable a precise assignment. The absence of sufficient resolution in the maps will therefore require biochemical experiments to decipher the mechanism of the described GTPases, which the authors may want to consider for a future submission.

ANSWER: We do not agree with the reviewer. Our density fits support our claims and the proteins that we used for comparison are homologs, which means that their active site is structured very similar to our GTPases. Moreover, we have phrased the text such that it expresses that our assigned nucleotide state is the most likely one but with some uncertainty (citation from the manuscript: ‘Both, GTPBP10 and GTPBP7, appear to have the non-hydrolysable nucleotide analogue Guanosine-5'-[(β,γ)-imido]triphosphate (GMPPNP) bound in their active sites (Supplementary Fig. 2).’). We

believe that we have been very open about shortcomings of our data and believe that we have provided sufficient evidence for our claims.

Second, uncertainty of experimental data is not a problem per se, but should be fully disclosed, be it at the level of resolution or B-factors. The authors seem to have used a limited range to map both B-factors and resolutions of their maps (c.f. Supplementary figures 14 & 15), which give the readers the impression that for example B-factors shown in supplementary figure 15 peak at 200 for intermediate 1 whereas the model of intermediate 1 provided for the first submission had B-factors ranging from 0 to more than 450.

ANSWER: The depiction we chose makes our data actually look worse than if we would have colour-coded over the entire B factor range. In our depiction, the lower confidence areas are much more obvious (see picture below). However, we can change the picture or adjust the labelling of the scale bar such that it is obvious that the magenta colour extends beyond B factor of 200 (for example '>=200'). The same holds true for the resolution range. Unfortunately, the reviewer makes his criticism sound like we try to hide anything here, which we do not intend to do.

Third, the notion that a large dataset is collected for multiple manuscripts but that it may be okay for these data to be withheld from the scientific community seems incompatible with the spirit of open science at Nature Communications. The response to a prior comment ("The authors should additionally upload their raw micrographs to EMPIAR to improve access for the entire scientific community"), which is: "We appreciate the suggestion to upload our data sets to EMPIAR and will consider it", does not suggest that the authors see that this is a real problem under the circumstances. Given the nature of the data and the fraction of the data that is part of this manuscript, publication of this manuscript in any journal should absolutely require prior acceptance of the raw data at EMPIAR to provide the scientific community with access to all of the underlying data.

ANSWER: This is a completely unfounded speculation from the reviewer and we do not believe that it justifies to reject our manuscript. We do not intend to publish an additional manuscript from this dataset as we have already made clear in our last revision response. The entire data will be made available upon request.

In summary, the data as presented in the current manuscript are not sufficient to support the claims of the authors.

Reviewer #2 (Remarks to the Author)

The authors have addressed appropriately the raised concerns.